# Inhibition of lysyl oxidases synergizes with 5-azacytidine to restore erythropoiesis in myelodysplastic and myeloid malignancies

Qingyu Xu[1], Alexander Streuer [1], Johann-Christoph Jann[1], Eva Altrock[1], Nanni Schmitt [1], Johanna Flach[1], Carla Sens-Albert[1], Felicitas Rapp [1], Julia Wolf[1], Verena Nowak[1], Nadine Weimer[1], Julia Oblander[1], Iris Palme[1], Mariia Kuzina [2], Ahmed Jawhar[3], Ali Darwich[3], Cleo-Aron Weis[4], Alexander Marx [4], Patrick Wuchter[5], Victor Costina[6], Evelyn Jäger[6], Elena Sperk [1], Michael Neumaier [6], Alice Fabarius[1], Georgia Metzgeroth[1], Florian Nolte[1], Laurenz Steiner[1], Pavel A. Levkin [2,7], Mohamad Jawhar[1], Wolf-Karsten Hofmann[1], Vladimir Riabov [1,8] ✉ & Daniel Nowak [1,8] ✉

Limited response rates and frequent relapses during standard of care with hypomethylating agents in myelodysplastic neoplasms (MN) require urgent improvement of this treatment indication. Here, by combining 5-azacytidine (5-AZA) with the pan-lysyl oxidase inhibitor PXS-5505, we demonstrate superior restoration of erythroid differentiation in hematopoietic stem and progenitor cells (HSPCs) of MN patients in 20/31 cases (65%) versus 9/31 cases (29%) treated with 5-AZA alone. This effect requires direct contact of HSPCs with bone marrow stroma components and is dependent on integrin signaling. We further confirm these results in vivo using a bone marrow niche-dependent MN xenograft model in female NSG mice, in which we additionally demonstrate an enforced reduction of dominant clones as well as significant attenuation of disease expansion and normalization of spleen sizes. Overall, these results lay out a strong pre-clinical rationale for efficacy of combination treatment of 5-AZA with PXS-5505 especially for anemic MN.

Myelodysplastic neoplasms (MN), including myelodysplastic syndromes (MDS), MDS/MPN overlap syndromes and there-of evolved leukemias are heterogeneous clonal diseases of hematopoietic stem and progenitor cells (HSPCs). Hypomethylating agents (HMAs) such as 5-Azacytidine (5-AZA) and decitabine are standard-of-care for higher-risk MDS and treatment options for MPN, such as primary myelofibrosis (PMF), MDS/MPN overlap syndromes, and chronic myelomonocytic leukemia (CMML)[1,2]. Although approximately 50% of MDS patients initially respond to HMAs, subsequent relapse is almost certain[3], therefore indicating an urgent need for compounds that synergize and further improve the beneficial effects of HMAs.

Although initially thought to be disorders that affect HSPCs, MN are now increasingly being recognized as diseases that also affect and interact with the bone marrow (BM) microenvironment[4,5]. Reciprocal

[1]Department of Hematology and Oncology, Medical Faculty Mannheim, Heidelberg University, Mannheim 68167, Germany. [2]Institute of Biological and Chemical Systems - Functional Molecular Systems, Karlsruhe Institute of Technology, Eggenstein-Leopoldshafen 76344, Germany. [3]Department of Orthopedic Surgery, Medical Faculty Mannheim, Heidelberg University, Mannheim 68167, Germany. [4]Institute of Pathology, Medical Faculty Mannheim, Heidelberg University, Mannheim 68167, Germany. [5]Institute of Transfusion Medicine and Immunology, German Red Cross Blood Service Baden-Württemberg-Hessen, Medical Faculty Mannheim, Heidelberg University, Mannheim 68167, Germany. [6]Institute of Clinical Chemistry, Medical Faculty Mannheim, Heidelberg University, Mannheim 68167, Germany. [7]Institute of Organic Chemistry, Karlsruhe Institute of Technology, Karlsruhe 76131, Germany. [8]These authors contributed equally: Vladimir Riabov, Daniel Nowak. ✉e-mail: vladimir.ryabov@medma.uni-heidelberg.de; daniel.nowak@medma.uni-heidelberg.de

interactions between HSPCs and aberrantly activated BM stroma were reported to contribute to MN progression[4,5]. In particular, mesenchymal stromal cells (MSCs) were identified as crucial niche components controlling HSPC renewal and differentiation in healthy and neoplastic hematopoiesis[6,7]. Aberrantly activated MSCs produce a variety of cytokines and growth factors, which may suppress erythropoiesis and facilitate deposition of aberrant stiff extracellular matrix (ECM)[8,9]. Interestingly, ECM stiffness and ligand composition were previously reported to control lineage commitment and differentiation of HSPCs[10,11]. Overall, current data suggest that targeting adverse factors of the BM microenvironment is a perspective therapeutic approach in hematological malignancies[4].

Lysyl oxidase (LOX) and LOX-like proteins 1-4 (LOXL1-4) are copper-dependent enzymes catalyzing extracellular cross-linking of collagen and elastin fibrils via lysine deamination. Overexpression of *LOX/LOXL* genes in PMF is associated with increased ECM deposition leading to BM fibrosis (BMF)[12-15]. Studies on solid tumors identified the involvement of LOX/LOXL proteins in the excessive deposition of collagen cross-linked ECM, tumor cell survival and chemoresistance[16-18]. Although LOX/LOXL enzymes have been proposed as potential therapeutic targets in solid tumors and indicated inferior outcome in acute myeloid leukemia[19], the functional roles of these enzymes and therapeutic potential in hematological malignancies are hardly explored.

In this study, we identify LOX/LOXL enzymes produced by MN-derived BM MSCs as emerging therapeutic targets. We show that the inhibition of LOX/LOXL enzymes synergizes with 5-AZA to restore erythroid differentiation of MN HSPCs in vitro and in vivo via BM stroma- and ECM-dependent mechanisms.

## Results

### Aberrant LOX/LOXL activity in the BM of MN patients can be inhibited by the pan-LOX/LOXL inhibitor PXS-5505

We previously reported an increased *LOXL2* gene expression in MDS MSCs[20]. In order to comprehensively characterize the role of *LOX/LOXL* genes in BM, we assessed their expression in different cellular compartments of primary BM in MN patients and healthy donors (HY) (Supplementary Tables 1 and 2).

Among the analyzed BM cell types, in-vitro expanded MN derived MSCs were the only relevant source of *LOXL2* expression (Fig. 1a) and other *LOX/LOXL* genes (Supplementary Fig. 1a). All *LOX/LOXL* genes, except for *LOXL1*, were significantly overexpressed in MN-derived MSCs as compared to HY MSCs (Fig. 1b). The overexpression of *LOX*, *LOXL2* and *LOXL3* was especially pronounced in CMML MSCs as compared to HY controls (Supplementary Fig. 1b). Overexpression of *LOX/LOXL* genes translated into significantly increased pan-LOX/LOXL activity in BM plasma (Fig. 1c). While LOX did not show relevant differences in concentration and only moderate increase of activity in MN (Supplementary Fig. 1c), LOXL2 displayed a significant increase of concentration and activity in MN BM plasma as compared to HY (Fig. 1d, e). Higher LOXL2 concentration and activity were related to higher BMF grading (Fig. 1f, g; Supplementary Table 3). There was no significant difference in LOX and LOXL2 concentration and activity between MDS and CMML patients (Supplementary Fig. 1d).

Due to the aberrant activity of LOX/LOXL enzymes in MN, we investigated the efficacy of the oral pan-LOX/LOXL inhibitor PXS-5505[21]. We found that PXS-5505 did not adversely affect the viability of MN MSCs and CD34 + HSPCs in a concentration range of 0.5-2 μM (Fig. 1h, i; Supplementary Table 4). In a time-course experiment, we determined that LOX/LOXL activity steadily accumulated in supernatants of MN MSCs during 48-hour culture, so that PXS-5505 was added every 24 h to achieve a continuous inhibition in vitro (Fig. 1j; Supplementary Table 5). At a dose of 2 μM, PXS-5505 significantly inhibited the pan-LOX/LOXL activity in supernatants of MSC culture (Fig. 1j). Functionally, the inhibitory effect of 2 μM PXS-5505 resulted in significantly decreased cross-linked collagen production (Fig. 1k).

Inhibition was fully comparable to a previously established pan-LOX/LOXL inhibitor, β-aminopropionitrile (BAPN)[21].

### Combination treatment with PXS-5505 + 5-AZA (P + A) synergistically increases erythropoietic differentiation of HSPCs in MN

In order to investigate the hypothesis that correction of elevated LOX/LOXL activity in the MN BM niche could be of therapeutic value, we studied the effects of PXS-5505 in autologous co-cultures of MSCs and CD34 + HSPCs followed by colony-forming unit (CFU) assays (Fig. 2a). In order to have a reference with an established drug known to be efficacious in the treatment of MN[22,23], we included 5-AZA monotherapy and also tested the combination of both drugs (P + A). Non-toxic active concentrations of PXS-5505 (2 μM) and 5-AZA (0.1 μM) in both HY and MN samples were determined using cell viability assays, recombinant human LOXL2 and LOXL3 activity assays and cross-linked collagen production inhibition assays (Fig. 1h, i; Supplementary Fig. 2a–e). PXS-5505 efficiently reduced collagen deposition by MSCs in MSC/HSPC co-cultures (Supplementary Fig. 2f). Moreover, only PXS-5505, but not 5-AZA, inhibited LOX/LOXL activity and collagen production in MSCs and MSC-derived fibroblasts (Fig. 1j, k; Supplementary Fig. 2d–g).

While the activity of the single substances 5-AZA and PXS-5505 was moderate, we observed a significant facilitation of erythroid colony production in CFU assays following combination treatment (P + A) (Fig. 2b). The induction of hemoglobin-containing progenitor cells was visually detectable in CFU plates and bulk cell pellets of the P + A treatment arm (Fig. 2c). Flow cytometry assessment of CFU bulk cells confirmed a clearly increased fraction of CD235a + CD45- erythropoietic progenitors after P + A treatment (Fig. 2d; Supplementary Fig. 3). In total, we performed flow cytometry assessment in n = 31 MN and n = 7 age-matched HY cases (Fig. 2e; Supplementary Table 6). Cases P10 and P23 were analyzed at two timepoints of BM aspiration. Erythropoietic response was defined as a significant increase of CD235a + CD45- cells in 5-AZA or PXS-5505 or P + A arms as compared to vehicle control (p < 0.05; Supplementary Fig. 4). According to this definition, n = 20 patients (65%) displayed an erythroid response to P + A (P1-P20, Supplementary Fig. 4a, b). Of these, n = 11 patients (35%) responded exclusively to P + A (P + A erythroid responders, P1-P11; Fig. 2e; Supplementary Fig. 4a). N = 9 patients (29%) also concomitantly responded to 5-AZA monotherapy (dual erythroid responders to 5-AZA and P + A, P12-P20; Fig. 2e; Supplementary Fig. 4b). The remaining n = 11 patients (P21-P31) were regarded as erythroid non-responders (Fig. 2e; Supplementary Fig. 4c). None of the tested conditions including P + A had any significant effects on the erythroid differentiation in HY HSPCs (Fig. 2e; Supplementary Fig. 4d). The comparison of P + A vs. 5-AZA monotherapy for improving erythroid response showed a relative risk (RR) of 0.45 with 95% confidence interval (95% CI) 0.24-0.80 (p = 0.0103). The assessment of fold changes in erythroid differentiation in n = 12 samples of P + A erythroid responders revealed synergistic effects of P + A as compared to the single substances (Fig. 2f). The effect of P + A treatment was associated with a predominant increase of the intermediate stage CD235a + CD71 + CD45- erythroid progenitors (Supplementary Fig. 5). Importantly, P + A treatment did not affect overall colony-forming potential of MN HSPCs in P + A erythroid responders (Supplementary Fig. 6a), but specifically increased erythroid colony output (Supplementary Fig. 6b).

The increase of erythroid cells was accompanied by a concomitant decrease in the percentage of immature CD33 + CD45 + myeloid progenitors in CFU assays (Fig. 2g). The analysis of megakaryocytic differentiation also revealed increased production of CD41 + megakaryocytes (MK) by P + A treatment (Fig. 2h; Supplementary Figs. 7 and 8). N = 14 of 26 MN samples (54%) treated with P + A displayed a megakaryocytic response. Of these, the co-cultured HSPCs

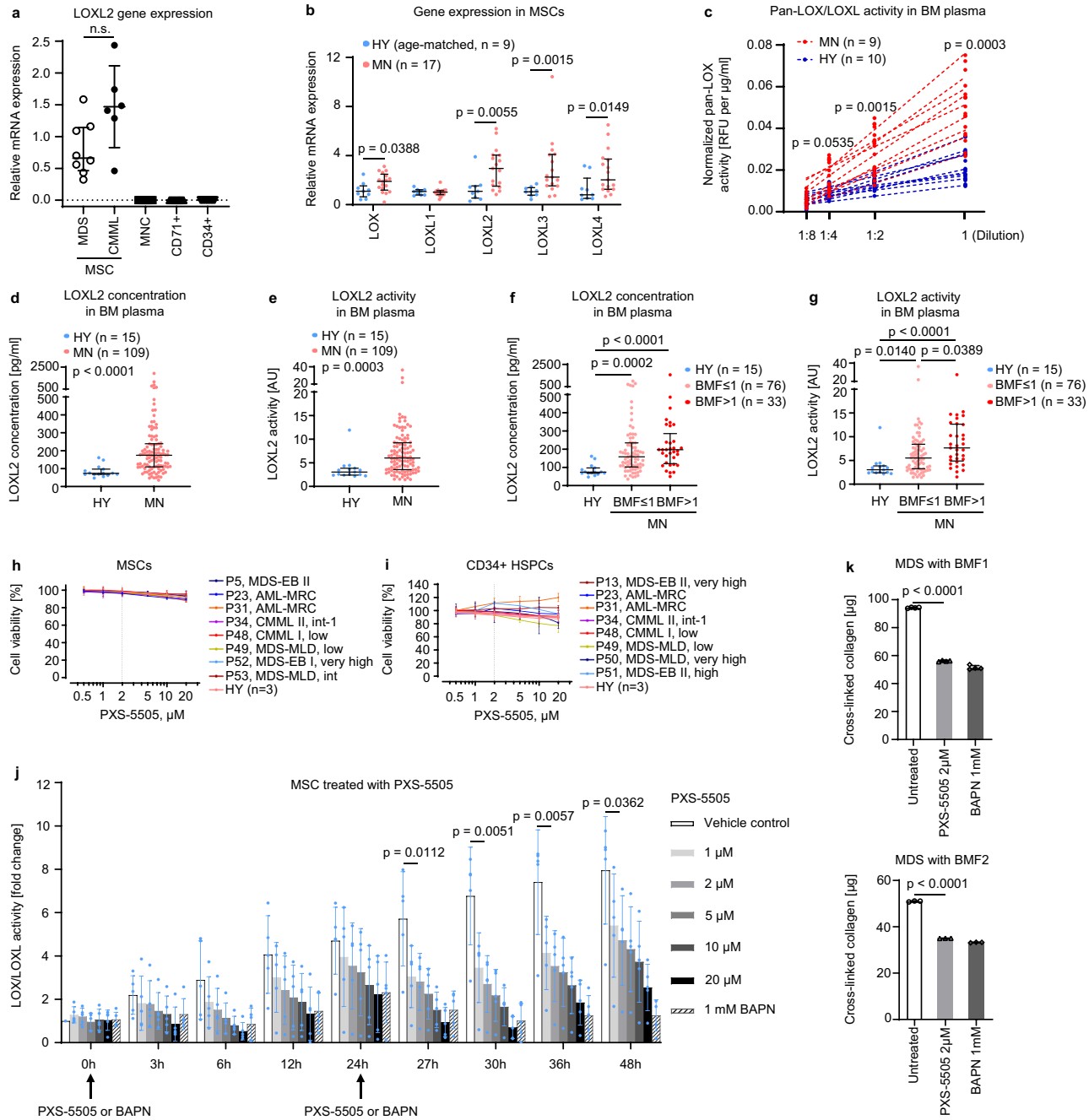

**Fig. 1 | Aberrant LOX/LOXL activity in the BM of MN patients can be inhibited by the pan-LOX/LOXL inhibitor PXS-5505. a** The *LOXL2* gene expression in BM cellular fractions of *n* = 20 MN patients (RT-qPCR), including *n* = 14 MSCs, *n* = 15 MNCs, *n* = 16 CD71 + cells and *n* = 12 CD34 + cells; median±interquartile ranges (IQR); Two-sided Mann-Whitney *U* test. **b** The *LOX/LOXL* gene expression in ex-vivo expanded MSCs (RT-qPCR). Individual data points for MN patients versus age-matched HY are presented; median±IQR; Two-sided Mann-Whitney *U* test. **c** Pan-LOX/LOXL activity was measured in BM plasma of MN and HY using Lysyl Oxidase Activity Assay Kit. The background level of pan-LOX/LOXL activity (signal-to-noise) was controlled by the treatment with a pan-LOX/LOXL inhibitor BAPN. The activity was normalized to the total protein content in BM plasma showing relative fluorescence units (RFU) per μg/ml; Two-sided Mann-Whitney *U*-test at different dilutions between MN and HY cohorts using mean data from duplicates for each dilution. **d, e** LOXL2 protein concentration and activity in HY (*n* = 15) vs MN patients (*n* = 109). The background level of LOXL2 activity (signal-to-noise) was controlled by BAPN treatment. **f, g** LOXL2 protein concentrations and activity are shown in MN patients stratified by bone marrow fibrosis (BMF) grade (BMF ≤ 1, *n* = 76 vs BMF > 1, *n* = 33). For **d**–**g**, the data are displayed as median ± IQR. Two-sided Mann-Whitney *U*-test (**d**, **e**) and Kruskal-Wallis test with Dunn's multiple comparisons (**f**, **g**) were performed. **h, i** BM MSCs (**h**) and HSPCs (**i**) from *n* = 11 MN and *n* = 3 HY were treated daily with the indicated concentrations of PXS-5505 for 11 days (MSCs) or 4 days (HSPCs), and cell viability was assessed using CellTiter-Glo assay; mean ± SD of 5 or 6 cell culture replicates. **j** Ex-vivo expanded MSCs were treated with indicated concentrations of PXS-5505 or BAPN every 24 h and pan-LOX/LOXL activity in supernatants of MSC culture was measured using the Lysyl Oxidase Activity Assay Kit. Combined data for *n* = 5 MN patients are presented as mean of fold changes ±SD; RM one-way ANOVA with Tukey's multiple comparisons for each time-point. **k** Production of cross-linked collagen was assessed in MSC-derived fibroblasts of *n* = 2 MDS-BMF samples treated for 7 days using PXS-5505 or BAPN. The data are mean of fibroblast culture quadruplicates (top) and triplicates (bottom)±SD; Ordinary one-way ANOVA with Tukey's multiple comparisons. *P* < 0.05 indicated statistical significance. Source data are provided as a Source Data file.

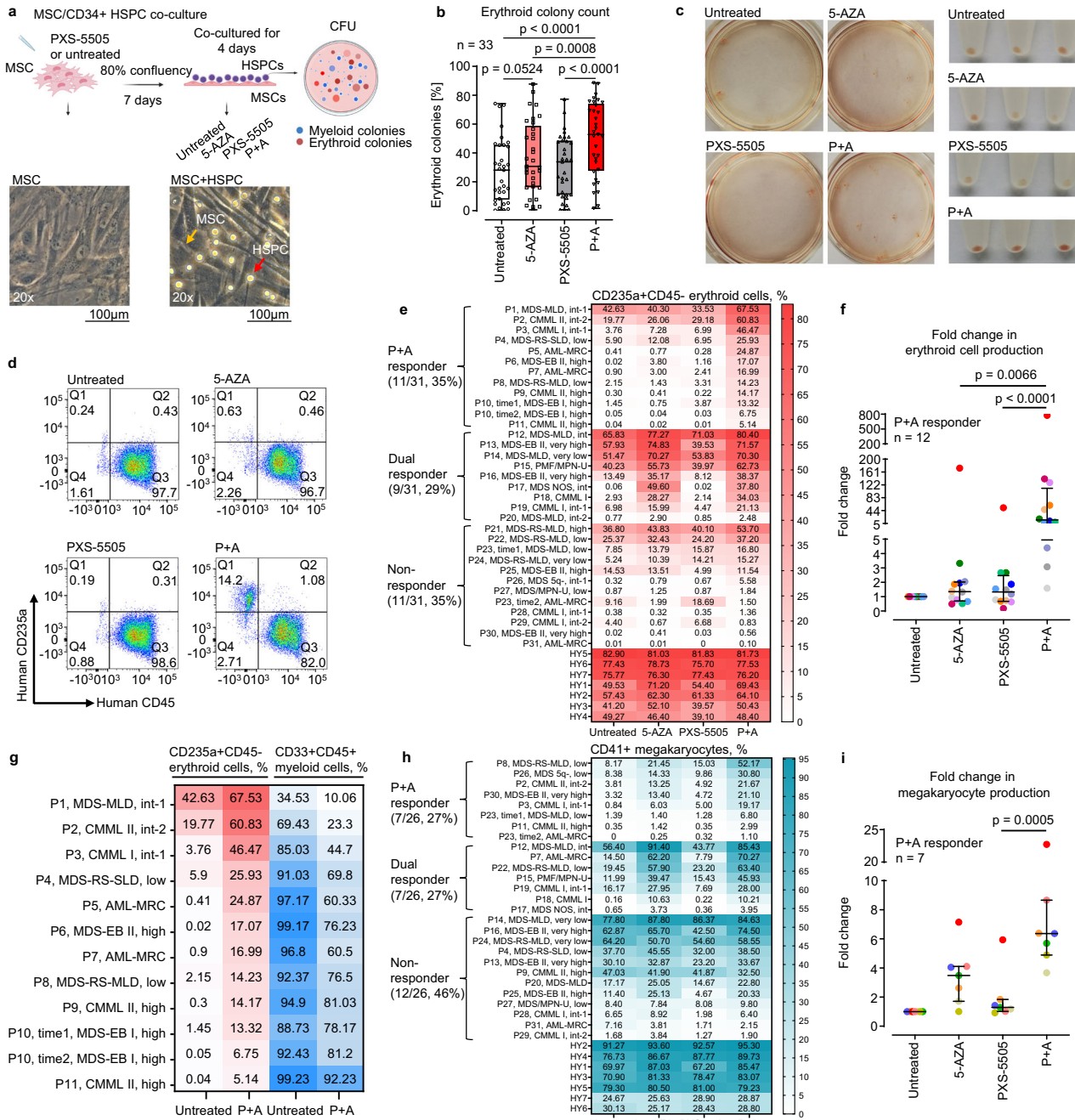

**Fig. 2 | PXS-5505 + 5-AZA (P + A) synergistically increases erythropoietic differentiation of HSPCs in MN. a** Experimental design of autologous MSC/CD34 + HSPC co-cultures. Schematic illustration was created using BioRender.com **b** HSPCs after co-cultures were re-plated on semisolid Methocult enriched methylcellulose-based medium with cytokines (Stemcell Technologies, H4435). The CFU assay results were presented as a percentage of erythroid colonies among total (erythroid + myeloid) colonies. Each box represents the IQR and median of the erythroid colony percentage in each group. Whiskers indicate Min and Max; median±IQR for a total of $n = 33$ samples (P10 and P23 were analyzed at two different BM aspiration timepoints); Friedman test with Dunn's multiple comparisons. **c** Examples of CFU assay plates containing hemoglobinized erythroid colonies and corresponding cell pellets are shown (P1). **d** CD235a + CD45- erythroid progenitors in CFU bulk cells were analyzed using flow cytometry. An example of sample (P9) is shown. **e** The percentages of CD235a + CD45- erythroid cells in CFU bulk cells were assessed by flow cytometry and presented as a heatmap. **f** Fold changes in the percentages of CD235a + CD45- cells compared to untreated co-cultures are shown for $n = 12$ P + A erythroid responder samples (P10 was analyzed at two different BM aspiration timepoints); median ± IQR; Friedman test with Dunn's multiple comparisons. **g** The changes in

the percentage of erythroid and myeloid cells between untreated and P + A-treated HSPCs in CFU assays are shown for $n = 12$ P + A erythroid responder samples. For **e, g**, the data are mean of independent co-culture triplicates or quadruplicates. **h** CD34 + HSPCs of $n = 26$ MN patients and $n = 7$ HY were co-cultured with autologous MSCs followed by the induction of megakaryocytic differentiation of CD34 + HSPCs for 21 days. The percentages of induced CD41 + megakaryocytes (MK) were assessed using flow cytometry and presented as a heatmap. The data are shown as a mean of independent co-culture duplicates or triplicates. **i** Fold changes in the percentages of CD41 + MK compared to untreated co-cultures are shown for $n = 7$ P + A megakaryocytic responders (for P23, time2 is not shown due to the absence of CD41 + cells in untreated arm); median±IQR; Friedman test with Dunn's multiple comparisons. $P < 0.05$ indicated statistical significance. MDS-MLD MDS with multilineage dysplasia, MDS-RS-SLD MDS with single lineage dysplasia and ring sideroblasts, AML-MRC acute myeloid leukemia with myelodysplasia-related changes, MDS-EB MDS with excess blasts, MDS-RS-MLD MDS with multiple lineage dysplasia and ring sideroblasts, PMF/MPN-U PMF/MPN-unclassifiable, MDS NOS MDS unclassified, MDS/MPN-U MDS/MPN-unclassifiable. Source data are provided as a Source Data file.

from $n = 7$ patients (27%) responded exclusively to P + A (P + A mega-karyocytic responders; Fig. 2h; Supplementary Fig. 8a), while $n = 7$ patient samples (27%) responded to 5-AZA monotherapy (dual mega-karyocytic responders; Fig. 2h; Supplementary Fig. 8b). P + A had no or only marginal effect on the megakaryocytic differentiation in HY HSPCs (Fig. 2h; Supplementary Fig. 8d). The comparison of P + A *versus* 5-AZA in megakaryocytic response showed a RR of 0.5 with 95% CI 0.24–0.99 ($p = 0.0889$). Of note, P2, P3, P8 and P11 were P + A responders in both erythroid and megakaryocytic assays (Fig. 2h, i).

Overall, this in-vitro data indicated that P + A synergistically restored erythropoiesis but also improved megakaryopoiesis and led to a reduction of immature myeloid cells in subgroups of MN patients.

## P + A facilitates erythroid differentiation of subclones with low mutational variant allele frequencies (VAFs)

In order to elucidate the clonal contribution to the augmented ery-thropoietic output in P + A-treated samples, we performed deep sequencing of input CD34 + HSPCs, post-CFU bulk cells and post-CFU sorted CD235a + CD45- progenitors from $n = 3$ P + A erythroid respon-ders (Supplementary Fig. 3 for sorting strategy). Interestingly, major recurrent mutations defining the dominant clone in the input HSPCs and bulk CFU outputs were either not present or had strongly reduced VAFs in the sorted CD235a + CD45- erythroid progenitors in all con-ditions (Fig. 3). This observation suggested that the increase in ery-thropoietic differentiation observed in P + A erythroid responders was largely facilitated by residual subclones with less severe mutational profiles.

## P + A improves erythropoiesis and reduces disease burden of MN in vivo

To interrogate whether beneficial effects of P + A were also ascertain-able in vivo, we utilized our previously established niche-based patient-derived xenograft (PDX) model of MN[20,24] based on NOD.Cg-*Prkdc*[scid] *Il2rg*[tm1Wjl]/SzJ (NSG) mice. The patient characteristics for in-vivo studies and experimental design are shown in Supplementary Table 6 and Fig. 4a, respectively.

As exemplified by case P32 with anemic PMF-BMF3 (Fig. 4b–d) and the summary of $n = 6$ MN cases (Fig. 4e, f), this model readily recapitulated an augmented efficacy of P + A treatment as opposed to vehicle or the single substances on clinically relevant disease parameters. Specifically, the P + A combination displayed the stron-gest inhibition of disease activity in the recipient animals as evi-denced by relative reduction of BM engraftment at endpoint analysis as compared to the initial baseline timepoint (Fig. 4b, e; Supple-mentary Fig. 9). As a further parameter of disease activity, we cal-culated the spleen indices of the PDX mice under treatment (Spleen index [%] = Spleen weight at endpoint/body weight*100). Spleen indices showed strongest containment of disease in the P + A con-dition (Fig. 4c, f). Specifically, the results presented in Fig. 4f indi-cated that 5-AZA alone expectedly induced a significant disease reduction and that this was slightly, but significantly augmented upon addition of PXS-5505 ($p = 0.0237$). PXS-5505 alone did not exert relevant reductions of BM engraftment or spleen sizes as compared to the untreated controls. However, apart from this, as a further disease activity indicator, we showed that our NSG PDX model was also able to reproduce BMF and hyperosteogeny by the transplanted PMF-BMF3 samples (e.g. P32). BMF formation was strongly inhibited by 5-AZA, PXS-5505 and P + A treatment (Fig. 4g, h). Of note, in contrast to 5-AZA, decreased BMF in PXS-5505-treated mice was observed despite comparably high BM engraftment (as visualized by intensity of human mitochondria staining), therefore validating its specific anti-fibrotic efficacy in vivo (Fig. 4g). The analysis of extracted human PDX samples by multi-timepoint whole exome sequencing and panel sequencing indicated a particularly strong clearance of dominant clones or subclones by P + A treatment. This

was evidenced by VAF reduction of *BCOR* and *RAD21* mutations in P32 (Fig. 4d; Supplementary Fig. 10a), *SRSF2*, *ASXL1* and *ETNK1* mutations in P18 (Fig. 4i) or *TET2*, *JAK2* and *ZRSR2* mutations in P33 (Fig. 4j). In P11, clonal composition remained stable throughout the experiment (Supplementary Fig. 10b).

Due to residual macrophage activity, the NSG model does not enable quantitative assessment of mature xenografted human ery-thropoiesis in peripheral blood[25]. However, improved human ery-throid differentiation in P11 and P32 as well as improved megakaryocytic differentiation in P11 was observed directly in vivo (Fig. 5a, b; Supplementary Fig. 11a, b). Moreover, augmented erythroid differentiation was also found in ex-vivo erythropoietin (EPO)-induced assays for P11, P3 and P30 (Fig. 5c–e; Supplementary Fig. 11c; Supple-mentary Fig. 12).

We also analyzed the impact of treatment conditions on the murine hematopoiesis of the PDX recipient NSG mice (Supplementary Figs. 13 and 14). The percentages of mouse erythroid, myeloid and megakaryocytic cells in BM were not affected by PXS-5505 and P + A (Supplementary Fig. 14a–c). Similarly, these treatment conditions did not significantly affect murine hemoglobin or platelet levels in per-ipheral blood (Supplementary Fig. 14d, e). Body weights as an indicator of animal well-being were not significantly perturbed throughout the 12 weeks of treatment (Supplementary Fig. 14f). As a further con-firmation of a favorable safety profile of PXS-5505, hematologic para-meters in healthy rats remained stable throughout a long term application of 70–140 mg/kg daily for 26 weeks (Supplemen-tary Fig. 15).

Finally, we assessed the clonal composition of sorted human CD45 + cells and human CD235a + CD45- erythroid progenitors from P11 and P30 PDX models. In contrast to the in-vitro data, erythroid cells of both patients were differentiated from clonally mutated progenitors with mutational VAFs comparable to the initial MN HSPCs (Supple-mentary Fig. 16).

Overall, these in-vivo studies showed that P + A not only induced superior erythroid production in human MN as compared to vehicle or the single substances but also achieved superior reduction of disease burden as evidenced by reduction of MN engraftment, spleen size, BM fibrosis and dominant cell clones.

## Induction of erythroid differentiation by P + A treatment is BM stroma-dependent

MSCs were previously shown to be potential therapeutic targets in MN[7,26]. Since LOX/LOXL inhibition putatively primarily targets collagen cross-linking of ECM in the BM niche, we asked whether erythroid differentiation induced by P + A is a result of MSC-mediated mechan-isms. First we compared the effect of P + A on MN HSPCs either cul-tured alone or together with autologous MSCs and found that the augmentation of erythroid differentiation was significantly increased by the co-culture with MSCs (Fig. 6a). Next, we asked whether ery-throid response was possibly dependent on intrinsic determinants of MSCs. Therefore, we co-cultured MSCs from P + A erythroid respon-ders with allogeneous HSPCs of patients who did not respond to P + A. In 3 out of 4 cases we observed notable increases of the erythroid differentiation of previously unresponsive HSPCs if cultured with allogeneous MSC from responders (Fig. 6b). Vice versa, co-culture of HSPCs of P + A erythroid responders with MSCs from erythroid non-responders reduced erythroid differentiation in 3 out of 4 cases (Fig. 6c). Collectively, these results suggested that P + A response was dependent on HSPC-MSC interaction.

## P + A-induced erythroid differentiation is dependent on integrin signaling and direct contact of HSPCs to BM stroma cells and ECM

Since we showed that synergistic effects of P + A treatment on MN erythropoiesis were dependent on the presence of MSCs, we were

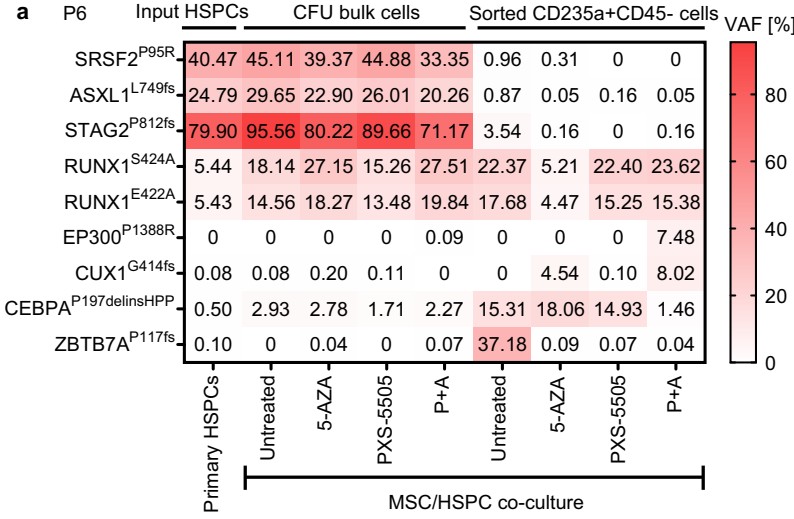

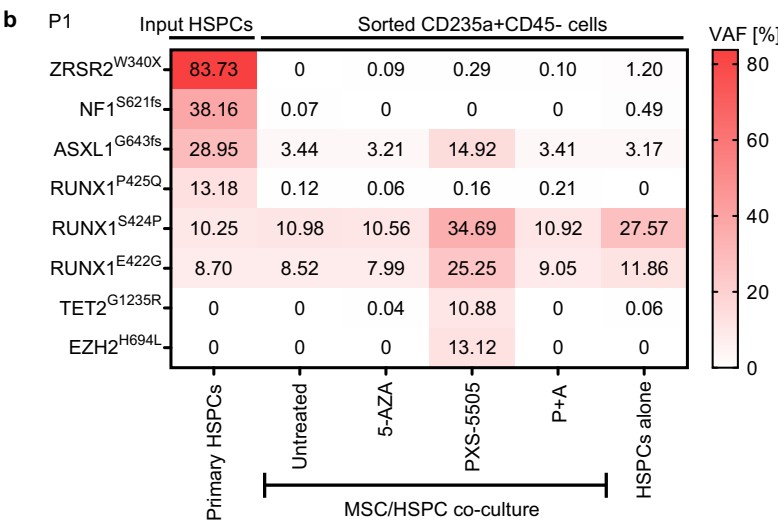

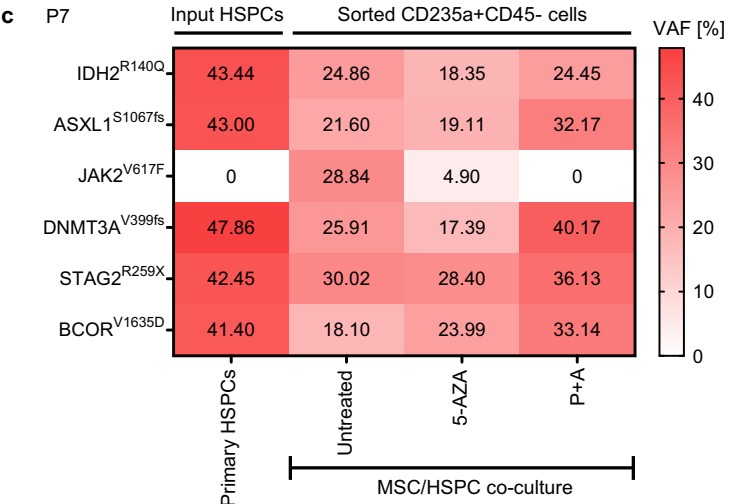

**Fig. 3 | P + A facilitates erythroid differentiation of subclones with low mutational variant allele frequencies (VAFs).** CD235 + CD45- erythroid progenitors of $n = 3$ P + A erythroid responders P6 (**a**), P1 (**b**) and P7 (**c**) were sorted from CFU assays post MSC/HSPC co-cultures and subjected to the panel sequencing in parallel with primary patient HSPCs. For P6, bulk CFU cells were also available for analysis. Heatmaps indicate VAFs (%) of mutations. Source data are provided as a Source Data file.

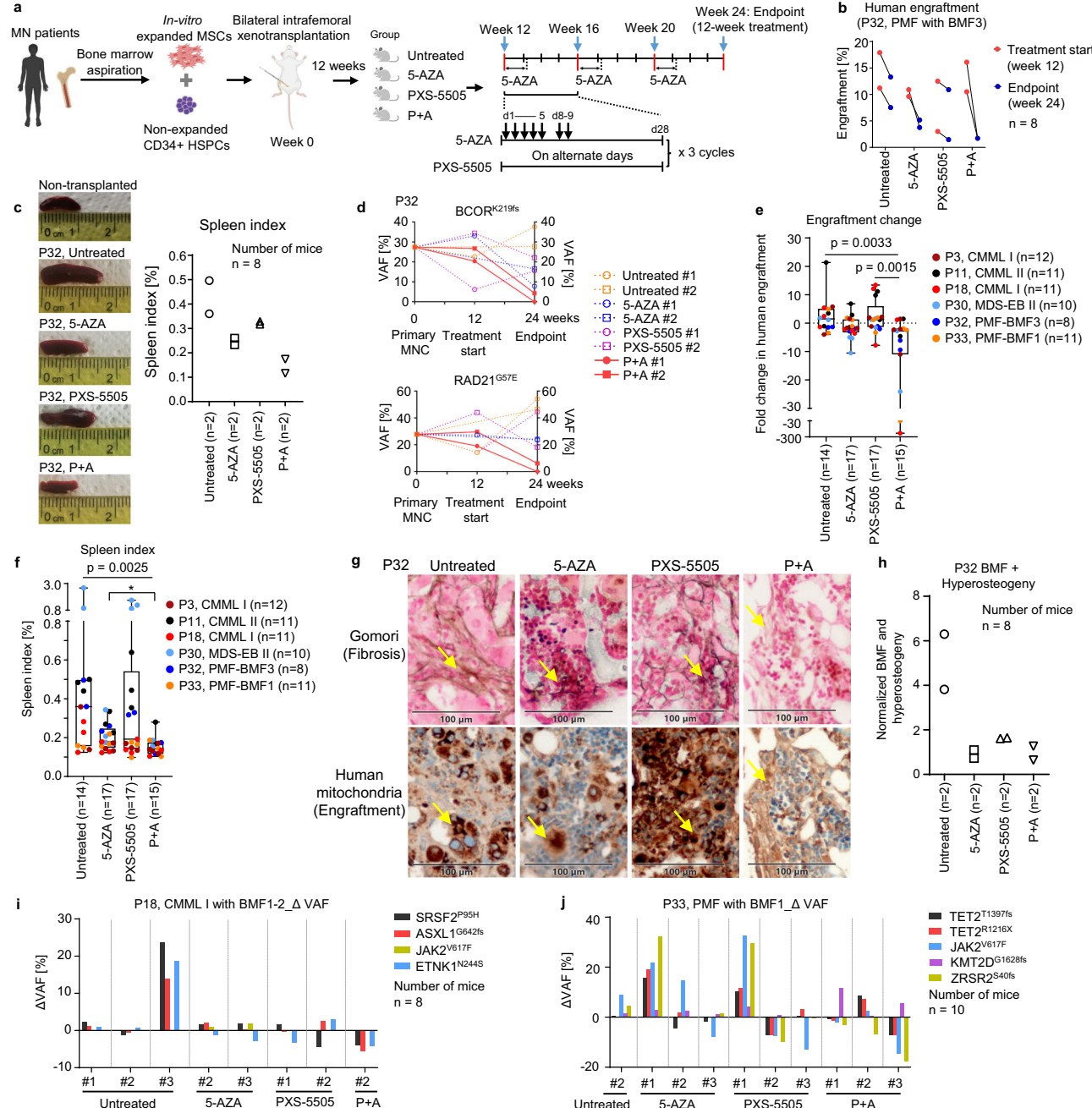

**Fig. 4 | Combination treatment with P + A reduces disease burden of MN in vivo.**
**a** PDX experimental design. Schematic illustration was created using BioRender.com. **b–d** PDX model of a PMF patient with BMF3 (P32). **b** Engraftment was assessed in the mouse BM at the treatment start (week 12) and endpoint (week 24) for *n* = 2 mice per treatment arm. **c** Representative photos of mouse spleens with corresponding spleen indices (*n* = 2 mice per treatment arm; Spleen index [%] = Spleen weight/body weight*100). **d** Dynamics of VAFs (%) for *BCOR* (K219fs) and *RAD21* (G57E) mutations in the course of treatment was assessed using WES of human CD45 + cells sorted from mouse BM. The data for individual mice are presented. **e** Summary of engraftment changes at the treatment start and endpoint (*n* = 63 PDX mice based on samples of *n* = 6 patients; Untreated *n* = 14, 5-AZA *n* = 17, PXS-5505 *n* = 17, P + A *n* = 15); Fold change (FC) = higher/lower engraftment value. In case of decrease in engraftment at the endpoint (week 24) compared to treatment start point (week 12), negative FC values were assigned. The data are median ± IQR. Kruskal-Wallis test with Dunn's multiple comparisons was used for statistical analysis. Each box represents the IQR and median of the engraftment changes in each group. Whiskers indicate Min and Max. **f** The comparison of spleen indices between untreated, 5-AZA, PXS-5505 and P + A treated mice at the endpoint.

The data are median ± IQR of *n* = 63 PDX mice (Untreated *n* = 14, 5-AZA *n* = 17, PXS-5505 *n* = 17, P + A *n* = 15) based on samples of *n* = 6 patients. Kruskal-Wallis test with Dunn's multiple comparisons was used for statistical analysis. Two-sided Mann-Whitney *U* test was used for calculating statistical significance between spleen indices of 5-AZA alone and P + A, *p* = 0.0237. Each box represents the IQR and median of the spleen index in each group. Whiskers indicate Min and Max. **g** Sequential sections of mouse femurs at the endpoint (P32) were stained using Gomori silver impregnation (left panel for each arm) or anti-human mitochondria antibody (right panel for each arm, visualized in brown). Arrows indicate reticular fibers and mitochondria. Scale bars: 100 μm. **h** Combined quantitative analysis of reticular fibers (BMF) and hyperosteogeny in mouse BM using QuPath software for P32. The data of *n* = 8 PDX mice (*n* = 2 per treatment arm) are presented as percentage of squares containing fibrotic + hyperosteogenic areas normalized to the engraftment rate (human mitochondria positivity in the mouse BM). **i, j** Delta (Δ) VAFs (VAFs [%] endpoint - VAFs [%] treatment start) for two patients (**i**: P18; **j**: P33); panel sequencing data for individual mice are presented. *P* < 0.05 indicated statistical significance. PMF-BMF, PMF with bone marrow fibrosis. Source data are provided as a Source Data file.

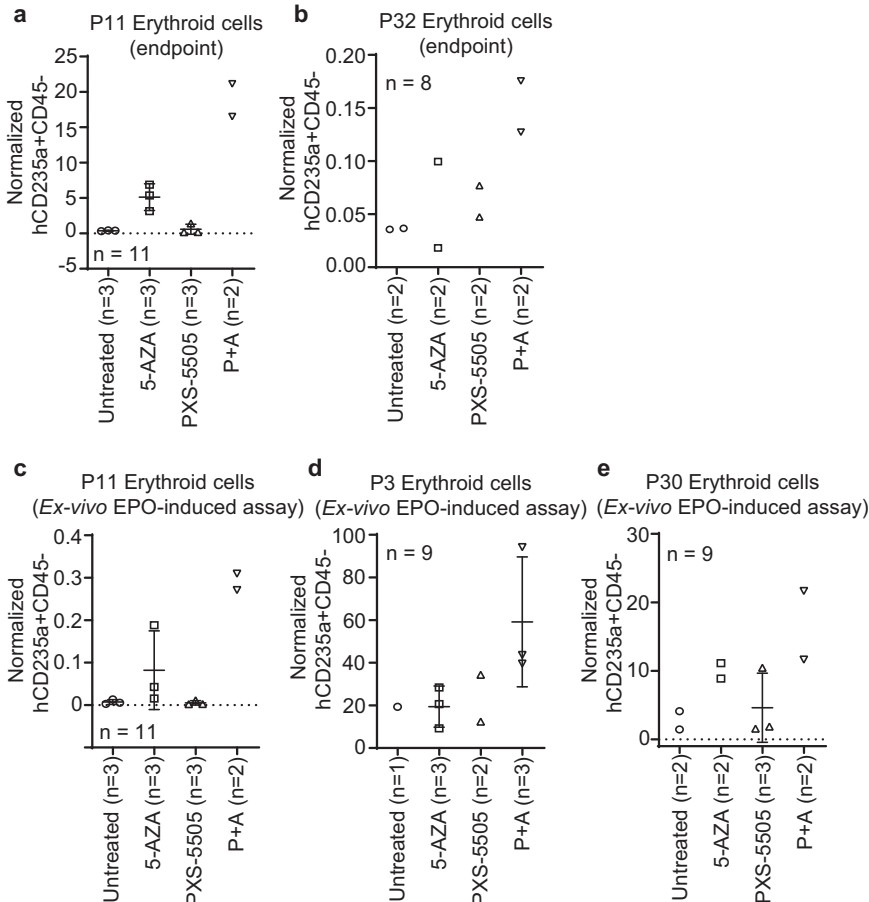

**Fig. 5 | Combination treatment with P + A facilitates erythroid differentiation in vivo. a, b** Percentages of human CD235a + CD45- erythroid progenitors at the endpoint (**a**: P11, *n* = 11 PDX mice [Untreated *n* = 3, 5-AZA *n* = 3, PXS-5505 *n* = 3, P + A *n* = 2]; **b**: P32, *n* = 8 mice; *n* = 2 mice per treatment arm) normalized to the percentage of human CD34 + CD33-CD71- HSPCs in the mouse BM (flow cytometry data). **c**–**e** EPO-induced assays for PDX of P11 (**c**, *n* = 11 mice [Untreated *n* = 3, 5-AZA

*n* = 3, PXS-5505 *n* = 3, P + A *n* = 2]), P3 (**d**, *n* = 9 mice [Untreated *n* = 1, 5-AZA *n* = 3, PXS-5505 *n* = 2, P + A *n* = 3]) and P30 (**e**, *n* = 9 mice [Untreated *n* = 2, 5-AZA *n* = 2, PXS-5505 *n* = 3, P + A *n* = 2]) at the PDX endpoint. Percentages of human CD235a + CD45- erythroid cells were normalized to the percentage of human CD34 + CD33-CD71- HSPCs inside sorted human CD45 + cells. The data are mean ± SD of individual mouse. EPO, erythropoietin. Source data are provided as a Source Data file.

interested whether this was mediated by soluble factors or direct cell-to-cell contact signaling. We therefore performed a transwell co-culture without direct MSC-to-HSPC contacts (Fig. 7a). As depicted in Fig. 7b and Supplementary Fig. 17a, b, the transwell co-culture abrogated the P + A-induced effects on erythropoiesis and favored myeloid differentiation (Supplementary Fig. 17c). Recovery of HSPCs in this experiment was similar in all studied conditions (Supplementary Fig. 17d). This data suggested a necessity for either direct cell-to-cell or cell-to-ECM contact for the observed effect in MN patient samples.

Of note, cultured MN MSCs produced robust layers of ECM (Fig. 7c). Microscopic evaluation of decellularized ECM revealed that PXS-5505 considerably reversed ECM morphology produced by MN MSCs to a phenotype more resembling the coarser fibrillar structure produced by HY MSCs (Fig. 7d; Supplementary Fig. 18; Supplementary Fig. 19a for case P3). These fibrils stained positively for collagen I (Fig. 7e). To test whether modified ECM alone was able to improve erythroid differentiation, we next cultured MN HSPCs of *n* = 3 P + A erythroid responders on decellularized ECM of either vehicle or PXS-5505 pre-treated autologous MSCs (Fig. 7f–h). Two of these patients (P3 and P11) were erythroid responders in both in-vitro co-culture and in the in-vivo xenografted model. Of note, HSPCs of these two patients showed improved erythroid response on decellularized ECM after P + A treatment of PXS-5505-treated MSCs (PXS-5505 ECM), albeit at slightly lower levels as compared to co-culture with viable MSCs (Fig. 7f, g).

Interestingly, mass spectrometry (MS)-based proteomic analysis revealed more similarities between P3 and P11 in terms of ECM changes after PXS-5505 treatment in MSCs as compared to P10 without PXS-5505-associated changes in ECM (Fig. 7i; Supplementary Fig. 19a for case P10). In particular, ECM changes in P3 and P11 involved enrichment in collagen type I alpha 2 chain (COL1A2) and TGF-β induced protein ig-h3 (βIG-H3), a known ligand for αVβ3 integrin[27], but also decreased abundance of collagen alpha-1(XI) chain (COBA1) and collagen-interacting protein lumican (LUM) (Fig. 7i, j; Supplementary Tables 7–9). All three cases showed reduced presence of secreted protein acidic and rich in cysteine (SPARC) in the ECM upon PXS-5505 treatment (Fig. 7i, j; Supplementary Tables 7–9).

We were next interested in the mechanism that enables differentiation of MN HSPCs on PXS-5505 ECM. It was previously shown that differential physical and biochemical ECM cues control lineage commitment of HSPCs via changes in cytoskeletal tension downstream of integrin signaling activation[11]. Therefore, we examined whether similar mechanism could be responsible for erythroid differentiation of patient-derived HSPCs in our decellularized ECM assays. We co-cultured HSPCs of P3 and P11 on autologous PXS-5505 ECM in the presence of myosin II inhibitor blebbistatin and rho-associated protein kinase (ROCK) inhibitor Y-27632 that regulate cytoskeletal tension downstream of integrin activation[11,28,29]. We observed complete inhibition of P + A-induced erythroid differentiation of HSPCs with concomitant increase of the myeloid cell differentiation in the presence of

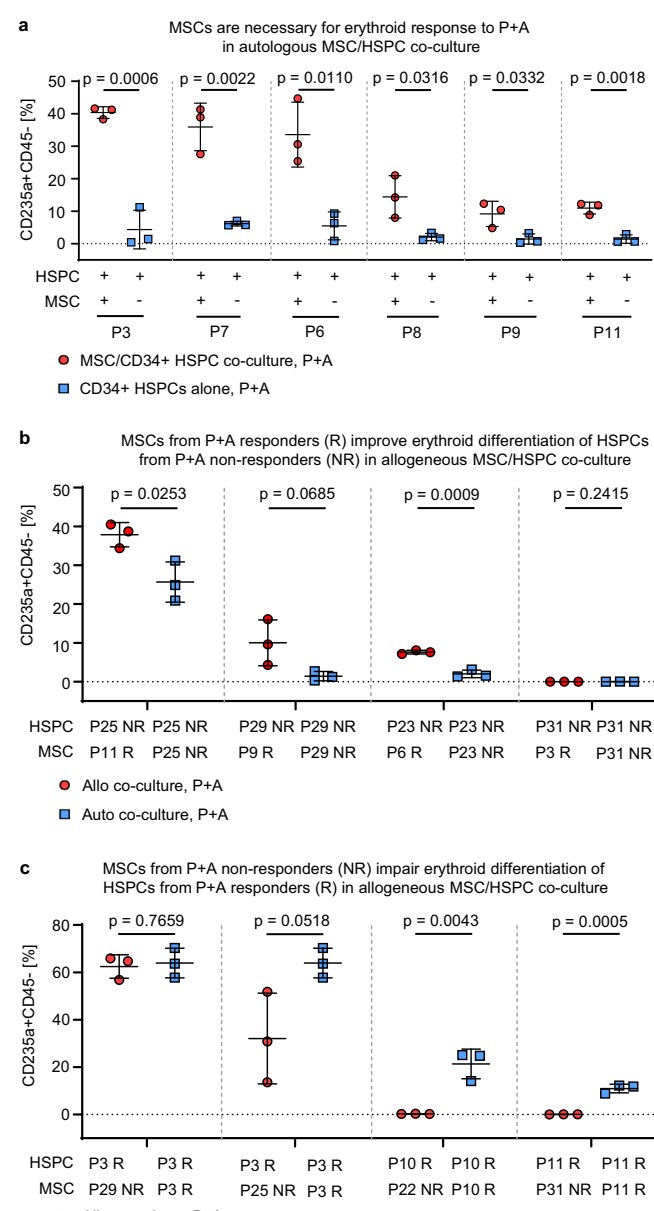

**Fig. 6 | Induction of erythroid differentiation by P + A treatment is BM stroma-dependent. a** CD34 + HSPCs from *n* = 6 P + A erythroid responders were treated with P + A either alone or in co-culture with autologous (auto) MSCs.
**b** CD34 + HSPCs from *n* = 4 erythroid non-responders (NR, P25, P29, P23 and P31) were co-cultured either with allogeneous (allo) MSCs from *n* = 4 P + A erythroid responders (R, P11, P9, P6 and P3) or auto MSCs. **c** CD34 + HSPCs from *n* = 3 P + A erythroid responders (R, P3, P10 and P11) were co-cultured either with allo MSCs from *n* = 4 erythroid non-responders (NR, P29, P25, P22 and P31) or auto MSCs. For **a**–**c**, the assessment of CD235a + CD45- erythroid progenitors was done after CFU assay using flow cytometry. The data are mean ± SD of co-culture triplicates. Statistical significance was analyzed using Two-sided unpaired Student's *t* test. *P* < 0.05 indicated statistical significance. Source data are provided as a Source Data file.

non-toxic concentrations of blebbistatin (2 μM, daily) and Y-27632 (5 μM, every 48 h) (Fig. 8; Supplementary Fig. 19b; Supplementary Fig. 20). Since αVβ3 integrin ligand βIG-H3 was enriched in PXS-5505 ECM, we assessed the role of αVβ3 integrin in ECM-mediated erythroid differentiation of MN HSPCs. Notably, an αVβ3 blocking antibody completely abrogated the effect of P + A on the erythroid differentiation of HSPCs cultured on PXS-5505 ECM (Fig. 8).

In summary, our data suggest that the PXS-5505-induced ECM restructuring influences αVβ3-mediated integrin signaling, which favors erythroid differentiation of MN HSPCs in the presence of 5-AZA.

## Discussion

Anemia and concomitant transfusion dependence are profound burdens for MN patients. In this study, we found that addition of the orally available pan-LOX/LOXL inhibitor PXS-5505 to standard of care treatment with 5-AZA considerably increased erythroid differentiation of primary MN-derived HSPCs in pre-clinical in-vitro and in-vivo models. Of note, this effect was frequently synergistic in heavily pre-treated MN cells that only displayed low-level response or were refractory to treatment with 5-AZA monotherapy.

The rationale to apply LOX/LOXL inhibitors in MN came from the recognition that these diseases re-program BM microenvironment in order to facilitate their own propagation[20]. To this end we and others had previously shown that upregulation of *LOX/LOXL* expression was a hallmark of MN MSCs[19,20,30]. Because increased LOX/LOXL activity may be one of the underlying mechanisms for BMF[12,13], which is a frequent finding in MN, we hypothesized that counteracting this dysregulation could either positively influence the course of disease on its own or improve efficacy of established treatment options for MN. Since LOX/LOXL expression and activity were increased in MSCs but not the diseased HSPCs, we performed in-vitro drug tests in autologous MSC/HSPC co-cultures. In these models we could show that PXS-5505 + 5-AZA significantly boosted erythropoiesis. The underlying mechanism was clearly dependent on interaction of HSPCs with MSCs and decellularized ECM. The changes produced by PXS-5505 in the ECM were characterized by alterations in the collagen network structure/alignment and ligand repertoire (e.g. re-distribution of collagen types and decreased lumican content). Remarkably, contact with this modified ECM alone was able to partly confer the P + A induced augmentation of erythropoiesis. By additional functional experiments, we showed that this effect was mediated by integrin/ECM interaction. This is consistent with observations in solid malignancies, where detrimental effects of increased *LOX/LOXL* expression were associated with the changes in ECM and integrin signaling, ultimately resulting in drug resistance[18]. Furthermore, a previous study reported that menatetrenone increased fibronectin production by MSCs and provided contact-dependent hematopoietic support to HSPCs via interaction with α4β1 integrin[31]. Our results highlight the importance of contact-dependent ECM/integrin interactions resulting in the restoration of erythroid differentiation in MN HSPCs. Importantly, we demonstrate therapeutic susceptibility of these interactions in MN using primary patient samples. Altogether, we and others suggest that treatments targeting clonally-mutated HSPCs alone may not be sufficient to restore their lineage differentiation and need to be combined with microenvironment-targeting compounds in order to alleviate cytopenia in MN.

Since PXS-5505 specifically targets MSCs and the ECM, the approach to combine 5-AZA and PXS-5505 is therefore an example of effective BM niche-targeted therapy for MN. Comparable proof of this concept is the TGF-β trap luspatercept[9,26], which received FDA approval for the treatment of very low- to intermediate-risk MDS in 2020 and was shown to increase hematopoietic support via restoration of the stromal cell-derived factor-1/C-X-C chemokine receptor type 4 axis between MSCs and HSPCs in MDS[26]. Interestingly, luspatercept-treated MDS MSCs preferentially supported healthy over malignant hematopoiesis[26]. In line with this study, we showed that erythroid progenitors derived from in-vitro MSC/HSPC co-cultures had strongly reduced mutational VAFs as compared to the initial patient HSPCs, suggesting that boosted erythropoiesis originated from minor sub-clones with intact erythroid differentiation capacity. Although the induction of erythropoiesis was the most obvious, P + A also had beneficial effects on other important clinical parameters such

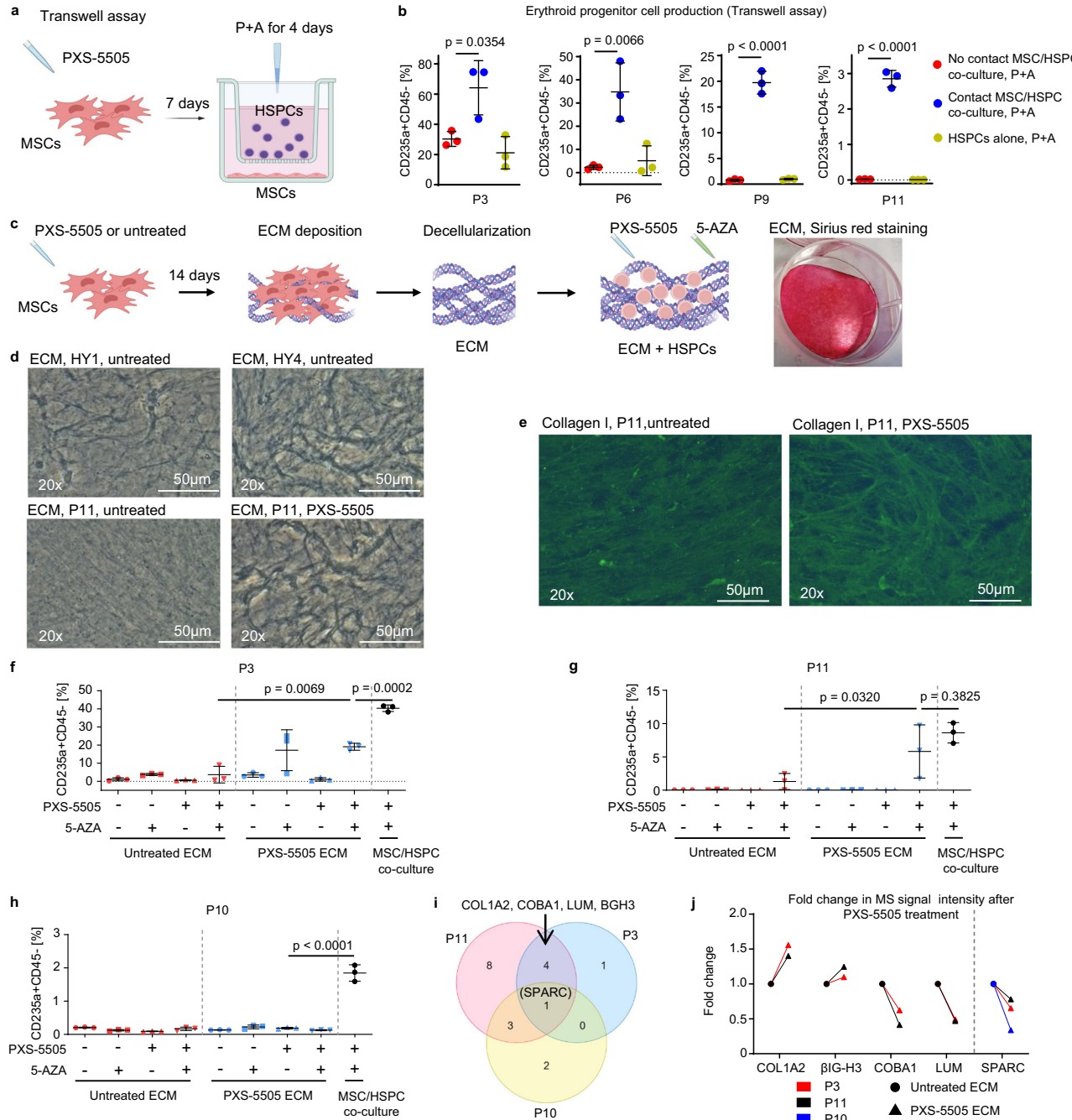

**Fig. 7 | P + A-induced erythroid differentiation in MN is dependent on active integrin signaling and direct contact of HSPCs to BM stroma cells and ECM.**
**a** Transwell co-culture experimental design. Schematic illustration was created using BioRender.com. **b** CD34 + HSPCs of *n* = 4 P + A erythroid responders (P3, P6, P9 and P11) were cultured alone or co-cultured with autologous MSCs in transwells or cell-cell contact conditions in the presence of P + A. Erythroid differentiation of CD34 + HSPCs was assessed using flow cytometry after CFU assay. The data are mean ± SD of independent co-culture triplicates. **c** Experimental design of decellularized ECM assay and exemplary image of deposited ECM stained using Sirius red. Schematic illustration was created using BioRender.com. **d** Phase contrast microscopy images of ECM deposited by healthy (HY1 and HY4) and MN MSCs (P11). The experiment was repeated three times with similar results.
**e** Immunofluorescence images of MN (P11) MSC-derived ECM stained with anti-collagen I antibody. The experiment was repeated twice with similar results.

**f**–**h** CD34 + HSPCs of *n* = 3 P + A erythroid responders (P3 [**f**], P11 [**g**] and P10 [**h**]) were cultured on the autologous untreated or PXS-5505 ECM. As a control, HSPCs were in co-culture with autologous MSCs and treated with P + A (black dots). Erythroid differentiation of HSPCs was assessed using flow cytometry after CFU assay. The data are shown as mean ± SD of independent co-culture triplicates. **i** MSC-derived ECM samples of P3, P10 and P11 were analyzed using mass spectrometry (MS). ECM proteins significantly affected by PXS-5505 treatment are presented by Venn diagram. **j** Fold changes in MS signal intensity for ECM proteins depicted in **i**. Statistical analysis for **b** and **f**–**h** was performed using ordinary one-way ANOVA with Tukey's multiple comparisons. *P* < 0.05 indicated statistical significance. ECM extracellular matrix, COL1A2 collagen type I alpha 2 chain, βIG-H3 TGF-β induced protein ig-h3, COBA1 collagen alpha-1(XI) chain, LUM collagen-interacting protein lumican, SPARC secreted protein acidic and rich in cysteine. Source data are provided as a Source Data file.

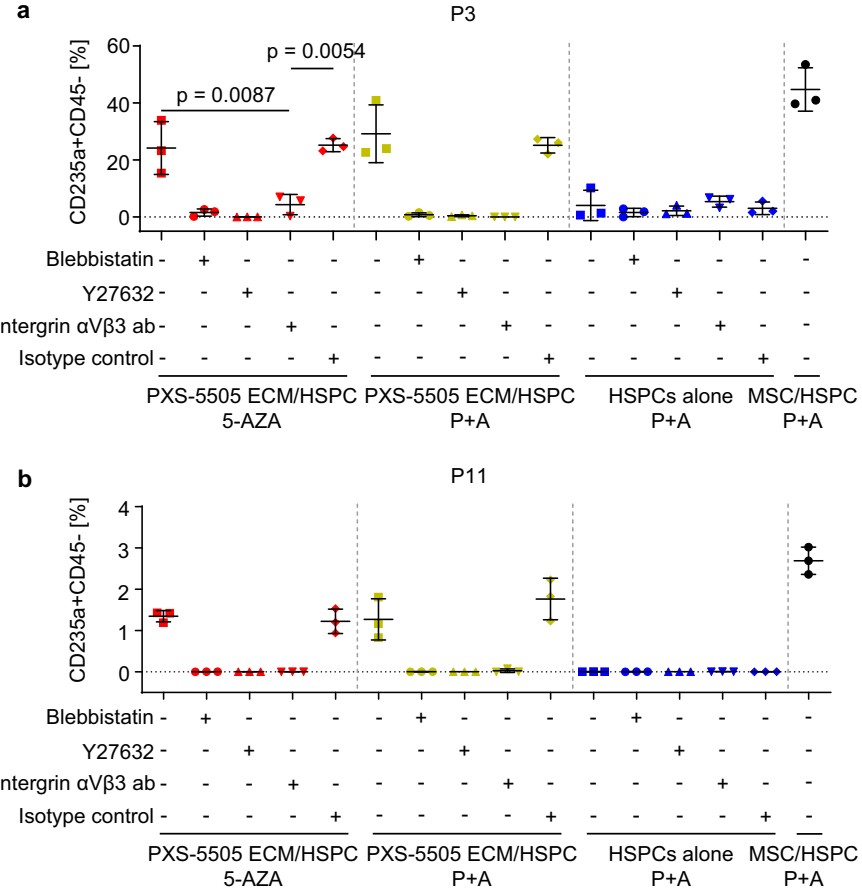

**Fig. 8 | P + A-induced erythroid differentiation in MN is dependent on integrin signaling.** CD34 + HSPCs of P3 (**a**) and P11 (**b**) were cultured on the PXS-5505 ECM in the presence of myosin II inhibitor blebbistatin, ROCK inhibitor Y27632 and anti-αVβ3 integrin antibody (ab) or corresponding isotype control. As controls, HSPCs were cultured alone (without MSCs or ECM) or co-cultured with autologous MSCs in the presence of P + A. Erythroid differentiation of HSPCs was assessed using flow cytometry after CFU assay. The data are mean ± SD of independent culture triplicates. Statistical analysis was performed using ordinary one-way ANOVA with Tukey's multiple comparisons. *P* < 0.05 indicated statistical significance. Source data are provided as a Source Data file.

as spleen size and improvement of megakaryocytic differentiation. It should be noted that in-vivo effects of P + A on the erythroid differentiation were heterogeneous. This reflects biological heterogeneity of MN and also suggests differential capability of HSPCs clones that populate the BM after P + A treatment for residual erythropoiesis.

The involvement of LOX in the regulation of megakaryocyte expansion was previously reported in mouse models[14,15]. However, our study highlights a different functional aspect of LOX/LOXL family in neoplastic megakaryopoiesis. In particular, our data using primary patient material suggest that the facilitation of de novo mega-karyopoiesis in HSPCs of some MN patients requires LOX inhibition with concomitant 5-AZA treatment.

The data of our current study were entirely obtained using primary patient material. Therefore, we provide an important pre-clinical foundation for the combination treatment of PXS-5505 and 5-AZA in anemic MN patients. The LOX/LOXL inhibitor PXS-5505 previously demonstrated a favorable safety profile in healthy human volunteers and is devoid of potentially hazardous off-target effects. Results of the underlying early phase clinical trial were reported in the Australian New Zealand Clinical Trials Registry, ACTRN12619000332123 and reviewed in Piasecki A et al.[32]. Moreover, our current study and other studies reported lack of PXS-5505 toxicity on hematopoiesis in mice and rats in both healthy and neoplastic conditions[15]. Taken together, based on the pre-clinical data presented here we are preparing an investigator-initiated phase I/II trial to evaluate the safety and

tolerability of the treatment of PXS-5505 + 5-AZA in MN patients sponsored by our institution.

## Methods

### Patient and healthy control samples for pre-clinical experiments

All patients and HY donors included in this study provided written informed consent and all interventions were performed in accordance with the Declaration of Helsinki. The study was approved by the Medical Ethics Committee II of the Medical Faculty Mannheim. Samples of MN patients were obtained from residual material from diagnostic BM aspirations. Patient characteristics are presented in Supplementary Tables 1, 3–6. Median age of the patients for in-vitro co-culture and in-vivo studies in mice was 73 years (range 49–87) with 70% males and 30% females (Supplementary Table 6). Hematopoietic cells from HY (≥50 years old) were obtained from bone specimen from femur endoprosthesis surgery (Supplementary Table 2). The BM of young HY (<50 years old) was collected by voluntary iliac crest puncture.

### Isolation of cellular fractions from BM material

Mononuclear cells (MNCs) were isolated using Ficoll-Paque (Cytiva, 17-5442-03) density gradient centrifugation. CD34 + HSPCs and CD71 + cell enrichment from MNCs were performed using MACS columns (Miltenyi Biotec, 130-046-703 for CD34 + HSPCs isolation, 130-046-201 for CD71 + cells enrichment). MSCs were selected by plastic adherence

and expanded ex vivo with StemMACS MSC expansion medium with XF supplement (StemMACS+XF medium, Miltenyi Biotec, 130-104-182). For in-vitro and in-vivo studies, expanded MSCs were used at passages 2 or 3. The phenotype of MSC (Lin$^{neg}$CD45$^{neg}$CD34-$^{neg}$CD271$^+$CD105$^+$CD73$^+$CD90$^+$) was validated by flow cytometry as previously described[33].

## Xenotransplantation experiments

All animals for xenotransplantation studies were housed under specific pathogen-free conditions in the animal facility of the Medical Faculty Mannheim, Heidelberg University at a 12 h/12 h day/night cycle in individually sterilized ventilated plastic cages with adjusted air temperature (21 °C) and 50% relative humidity. PDX models were established in 8-10 week-old NSG female mice (Jackson Laboratory, 005557) as reported previously[24]. Female NSG mice were used in the study since they are known to show higher success rate of obtaining human engraftment compared to males[34]. In modification to the previous protocol, mice were sublethally conditioned with Busulfan (Sigma-Aldrich, B2635; 25 mg/kg, intraperitoneal injection [i.p.]) 48 h and 24 h before intrafemoral injection of cell suspension mixture containing non-expanded BM CD34 + HSPCs ($1 \times 10^5$ per femur, injection in both femurs) and in-vitro expanded patient-autologous MSCs ($5 \times 10^5$ per femur, injection in both femurs). These MSCs were not pre-treated before injection into NSG mice. Long-term engraftment in the BM was assessed at 12 weeks post-transplant. Positive human engraftment was defined as >0.5% of human CD45 + cells among human and murine CD45 + cells in the murine BM. Engrafted mice were treated with vehicle, 5-AZA (Sigma-Aldrich, A2385, 1.7 mg/kg, subcutaneous injection in water), PXS-5505 (Pharmaxis Ltd, 30 mg/kg, intraperitoneal injection in olive oil), or P + A. Mice received 3 cycles of 5-AZA according to the "5 + 2" schedule (once daily; 5-day treatment, 2-day break, 2-day treatment followed by 21-day break). PXS-5505 was administered every 48 h. The following termination criteria were applied for PDX experiments: body weight loss >20% from baseline, apathy, difficulty moving and signs of shortness of breath. At the endpoint, the bones (tibia, femur and ilium) were crushed for BM extraction followed by flow cytometry and fluorescence-activated cell sorting (FACS). Fold changes (FC) in engraftment at the endpoint compared to the treatment start were calculated as follows: FC = higher engraftment value/lower engraftment value. In case of increase in engraftment at the endpoint compared to treatment start point, positive FC values were assigned. In case of decrease in engraftment at the endpoint, negative FC values were assigned. In total, $n = 63$ xenografts from $n = 6$ patients were established in this study. Animal experiments were performed in accordance with institutional guidelines and approved by the state authority Karlsruhe, Germany (G164-18).

## PXS-5505 toxicity studies in rats

Wistar Han rats were housed in polycarbonate shoebox cages with corn cob bedding and 12-hour light/12-hour dark cycle. The temperature was set to be maintained between 20–26 °C. The humidity was maintained between 40–70%. Rats were fed *ad libitum* with rodent diet and had a free access to the sterilized water. Rats (males and females between 28–32 weeks old) were treated daily with 70 mg/kg, 100 mg/kg and 140 mg/kg of PXS-5505 using oral gavage for 26 weeks ($n = 15/15$ females/males per each treatment arm). The hematological parameters were assessed at the end of the study. The study was done in compliance with the Organization of Economic Cooperation and Development (OECD) Principles on Good Laboratory Practice ENV/MC/CHEM (98) 17 (Revised in 1997, Issued January 1998) and the United States Food and Drug Administration (US FDA) Good Laboratory Practice (GLP) Regulations for Nonclinical Laboratory Studies (21 CFR Part 58) guidelines at Pharmaron, China. The study was approved

by the Institutional Animal Care and Use Committee (IACUC) of Pharmaron, China (IACUC approval number 19-215).

## RNA isolation, cDNA synthesis and quantitative reverse transcription PCR (RT-qPCR)

RNA was isolated using the AllPrep DNA/RNA mini or micro Kit (Qiagen, 80204 or 80284 respectively). cDNA synthesis was performed using the QuantiTect Reverse Transcription kit (Qiagen, 205311). RT-qPCR for the analysis of *LOX* and *LOXL1-4* gene expression was performed using a LightCycler 480 Instrument II (Roche Life Science) and LightCycler 480 SYBR Green I Master mix (Roche Life Science, 4887352001). Gene expression data were normalized to the expression of β-glucuronidase (*GUS*). Relative gene expression were calculated using the $2^{-\Delta\Delta Ct}$ method[35]. Primer sequences are summarized as below:

*GUS*:
Forward primer (5′– 3′), gaaaatatgtggttggagagctc
Reverse primer (5′– 3′), ccgagtgaagatccccttttta
*LOX*:
Forward primer (5′– 3′), ggatatagcggtacatatgatccttag
Reverse primer (5′– 3′), aagctctgtctgtattgtctactggt
*LOXL1*:
Forward primer (5′– 3′), cgctatgcatgcacctctc
Reverse primer (5′– 3′), gatgtccgcattgtaggtgtc
*LOXL2*:
Forward primer (5′– 3′), ggagaggacatacaataccaaagtgt
Reverse primer (5′– 3′), ccatggagaatggccagtag
*LOXL3*:
Forward primer (5′– 3′), acaggctggacccacagt
Reverse primer (5′– 3′), ctgcagctcaagttgtccag
*LOXL4*:
Forward primer (5′– 3′), gctgcacaactgccacac
Reverse primer (5′– 3′), ggttgttcctgagacgctgt

## CellTiter-Glo (CTG) cell viability assay

Patient MSCs were cultured in StemMACS+XF medium. BM CD34 + HSPCs were cultured in StemSpan SFEM II medium (Stemcell Technologies, 09655) supplemented with HSPC expansion mix, containing recombinant human (rh) stem cell factor (rhSCF, Stemcell Technologies, 78062, 50 ng/ml), rh fibroblast growth factor-1 (Thermofisher scientific, 13241-013, 10 ng/ml), rh fms-like tyrosine kinase 3/fetal liver kinase-2 ligand (Stemcell Technologies, 78137, 50 ng/ml) and rh thrombopoietin (Stemcell Technologies, 78210, 10 ng/ml). The cells were seeded at a concentration of $5 \times 10^4$ cells/ml for CD34 + HSPCs and $6 \times 10^4$ cells/ml for MSCs in 48-well plates (150 μl/well) and treated using PXS-5505 or 5-AZA as described in figures and figure legends for individual experiments. Both PXS-5505 and 5-AZA were dissolved in sterile water. At the end of treatment, 150 μl of cell suspension were mixed with 150 μl of CTG reagent (Promega, G7571) and incubated for 2 min at room temperature (RT) with shaking to ensure cell lysis. The lysates were transferred into 96-well half-area white microplates (Greiner bio-one, 675074) and luminescent signal was measured using Tecan Infinite F200 Pro Microplate Reader.

## Pan-LOX/LOXL activity detection and inhibition assays

To obtain BM plasma from patients and HY, BM samples were spun down for 5 min, $400 \times g$, RT. Supernatants were centrifuged again for 10 min, $3000 \times g$, RT and stored at −80 °C. The assessment of pan-LOX/LOXL activity was performed using the Lysyl Oxidase Activity Assay Kit (Fluorometric) (Abcam, ab112139) according to the manufacturer's instructions. The background level of LOX and LOXL2 activity (signal-to-noise) was controlled by BAPN treatment.

MSCs were cultured in 12-well plates until 80% confluency in StemMACS+XF medium. The medium was changed for RPMI 1640 without phenol red (Gibco, 11835-063), 1 ml/well. After 24 h of culture,

PBS (vehicle control) or PXS-5505 or 5-AZA or BAPN (TCI, A0796) was added at indicated concentrations and supernatants were harvested after 0 h, 3 h, 6 h, 12 h and 24 h. Cells were treated with inhibitors again and supernatants were harvested at 27 h, 30 h, 36 h and 48 h for pan-LOX/LOXL activity measurement using the Lysyl Oxidase Activity Assay Kit (Fluorometric).

Rh LOXL2 and rhLOXL3 enzymes (R&D Systems, rhLOXL2: 2639-AO-010, rhLOXL3: 6069-AO-010) were treated with PXS-5505 or 5-AZA at indicated concentrations. LOX/LOXL activity was measured after 40 min of incubation at 37 °C using the Lysyl Oxidase Activity Assay Kit (Fluorometric).

Fluorescent signal was detected using the following parameters in Tecan Infinite F200 Pro Microplate Reader: Excitation/Emission = 535/590 nm.

### Assessment of LOX and LOXL2 concentration and activity in BM plasma

Simoa bead technology was used to determine concentrations and activity of LOX and LOXL2 enzymes in BM plasma samples as described previously[36]. Specifically, LOX or LOXL2 capturing antibodies were used in combination with activity-based probes to measure activity. The background level of LOX and LOXL2 activity (signal-to-noise) was controlled by BAPN treatment. Alternatively, secondary anti-LOX or anti-LOXL2 antibodies were used to measure protein concentration.

### Collagen production assessment in MSC-derived fibroblasts and MSCs after co-culture with HSPCs

MSCs were cultured in StemMACS+XF medium in the presence of 100 ng/ml connective tissue growth factor (rh CTGF, Peprotech, 120-19) and 50 μg/ml L-ascorbic acid (Sigma-Aldrich, A92902) for 14 days to induce fibroblast differentiation as described previously[37]. During differentiation, medium was changed every 3 days. Fibroblasts were cultured in 100 mm dishes in StemMACS+XF medium until 80% confluency. Medium was changed for StemMACS+XF supplemented with LOX/LOXL-containing supernatants harvested from the corresponding MSCs after 4 days of culture to facilitate collagen cross-linking. Fibroblasts were cultured for additional 7 days in the presence of PXS-5505, 5-AZA and PXS-5505 + 5-AZA (P + A) at indicated concentrations or left untreated. Drugs were added daily and medium changed on day 4.

For collagen production in MSC/HSPC co-culture, MSCs were co-cultured with HSPCs for 4 days with daily treatments of 5-AZA, PXS-5505 and P + A. HSPCs were removed from MSCs by pipetting before assessment of a collagen production.

Both fibroblasts and MSCs after treatment were washed in PBS, treated with 1 ml of 0.5 M acetic acid (Sigma-Aldrich, 1.00063), 0.1 mg/ml pepsin (Sigma-Aldrich, P7000), scraped off the plastic surface and rocked overnight at 4 °C. Pepsin digested samples were sheared using a 26 G needle. Acid neutralising reagent (100 μl) from Sircol Soluble Collagen Assay Kit (Biocolor, S1000) was added and samples were spun down at 10,000 × g for 10 min, 4 °C. 100 μl of 1% Brij97 (Sigma-Aldrich, P6136) solution with 25 mM Hepes, 150 mM NaCl, 5 mM MgCl$_2$ (PH 7.5) was added to the pelleted insoluble collagen fraction and incubated for 2–3 h at 65 °C with shaking. Samples were spun down at 10,000xg for 10 min, 4 °C and supernatants containing solubilized cross-linked collagen were collected. Collagen content was measured using Sircol Soluble Collagen Assay Kit according to the manufacturer instructions.

### MSC/CD34 + HSPC co-culture experiments and CFU assays

For co-culture experiments, MSCs ($2 \times 10^4$/ml) were cultured in 24-well plates in 1 ml StemMACS+XF medium and treated daily with 2 μM PXS-5505 for 7 days. Monolayers were co-cultured with autologous or allogeneic CD34 + HSPCs ($2 \times 10^4$) in 1 ml StemSpan SFEM II medium with human HSPC expansion mix. Co-cultures were treated daily for

4 days with 5-AZA (100 nM), PXS-5505 (2 μM), P + A or vehicle (PBS). HSPCs were gently separated from co-culture and seeded into semi-solid MethoCult H4435 Enriched Methylcellulose-based medium with human recombinant cytokines (Stemcell Technologies, H4435) for CFU assays (3000 cells/35 mm dish, 1 ml). Colonies were counted manually with a Leica DMi1 inverted light microscope (Leica Microsystems) after 14-day incubation. CFU bulk cells were analyzed by flow cytometry. For transwell co-cultures, MSCs were treated daily with 2 μM PXS-5505 for 7 days in 24-well plates (Corning Falcon, 353504). The StemMACS+XF medium for MSC cultures was changed for StemSpan SFEM II medium with human HSPC expansion mix, and $1 \times 10^4$ autologous CD34 + HSPCs were cultured in the 0.4 μm cell culture inserts (Corning Falcon, 353495) above MSC monolayers. The drugs were added daily for 4 days followed by CFU assays.

### Megakaryocytic differentiation assay

HSPCs from MSC/CD34 + HSPC co-culture assay post treatment were harvested and seeded in StemSpan SFEM II medium containing StemSpan Megakaryocyte Expansion Supplement (Stem Cell Technologies, 02696) at a concentration of $1 \times 10^4$ cells/ml. Medium was changed every 7 days. After 21 days of incubation, megakaryocytic differentiation was assessed using flow cytometry.

### Ex-vivo EPO-induced assay

Human CD45 + cells sorted from the mouse BM at the endpoint of xenograft experiments were cultured for 21 days in StemSpan SFEM II medium supplemented with rh EPO (Stemcell Technologies, 78007, 5U/ml), rh interleukin 3 (Promocell, C-61321, 5 ng/ml) and rh stem cell factor (100 ng/ml). Erythroid differentiation was assessed using flow cytometry.

### Flow cytometry and cell sorting

The following fluorescent antibodies were used for cell surface staining: anti-mouse Ter119-APC (Clone TER-119, eBioscience, 17-5921-82, 1:20), anti-mouse Gr-1-BV786 (Clone RB6-8C5, BD Biosciences, 740850; 1:20), anti-mouse/human CD11b-APC (Clone M1/70, Biolegend, 101212; 1:100), anti-mouse CD41-APC (Clone MWReg30, Biolegend, 133914; 1:20), anti-mouse CD45-APC-Cy7 (Clone 30-F11, Biolegend, 103116, 1:100), anti-human CD45-PE (Clone HI30, BD Bioscience, 555483, 1:100), anti-human CD235a-PerCP-Cy5.5 (Clone HI264; Biolegend, 349110; 1:40 for in-vitro assays, 1:10 for in-vivo assays), anti-human CD71-PE-Cy7 (Clone CY1G4; Biolegend, 334112, 1:5000), anti-human CD41-PE-Cy7 (Clone HIP8; Biolegend, 303718, 1:5000), anti-human CD33-APC (Clone WM53; Biolegend, 303408, 1:100), anti-human CD34-FITC (Clone 561; Biolegend, 343604, 1:100). Fc receptors were blocked using anti-human (1:10) ± anti-mouse (1:20) Fc blocking reagent (Miltenyi biotec, human 130-059-901; mouse 130-092-575) and cells were labeled with antibodies for 30 min at 4 °C in FACS buffer (PBS with 0.4% BSA and 0.02% NaN$_3$). Cells were washed once using 1.5 ml BD CellWASH (BD Biosciences, 349524) and resuspended in FACS buffer containing SYTOX blue (Thermofisher Scientific, S34857, 1:2000). Samples were acquired using a BD FACSMelody, BD FACSAria IIu and FACSAria™ Fusion Cell Sorters. Human CD45 + and CD235a + CD45- cells from murine PDX models or in-vitro CFU assays were sorted and used for functional assays or lysed in RLT + buffer (Qiagen, 1053393) for downstream molecular analyses. Flow cytometry data were analyzed using the FlowJo software (version 10.5.3).

### Whole exome sequencing (WES)

WES was performed as described previously[38]. In details, DNA was isolated using AllPrep DNA/RNA Kit (Qiagen, 80204). For WES, 250 ng of high molecular weight genomic DNA from patient and murine PDX samples was subjected to the Nextera DNA Flex Kit (Illumina, 20025524) and subsequent hybrid capture using the xGen Exome

Research Panel (IDT). The final library pools were sequenced on a S4 NovaSeq Flow Cell with 150 bp paired end reads. For mutational calling, raw sequencing data were subjected to a previously established bioinformatical pipeline[39] adapted for the analysis of xenograft samples, including fastq trimming trimmomatic version 0.39, alignment bwa version 0.7.9 and PCR deduplication using MarkDuplicates (Picard) version 2.20.5. BAM files were re-aligned and recalibrated with the gatk bundle version 3.8. Potentially overlapping forward and reverse reads were soft clipped using bamUtil clipOverlap (version 1.0.14). After mapping to hg 19, somatic mutations were called by gatk Mutect 2 gatk version 4.1.3.0. DNA from matched MSCs served as germline controls. Only variants passing FilterMutectCalls were further annotated with annovar 3 and subsequently verified by manual visualization in IG Viewer (version 2.3.98). For homogenous comparison among multiple PDX from a single patient, each tumor normal comparison was called independently in this manner. Subsequently, all passed filter sites from all PDX were combined using gatk genotypeMergeOptions UNIQUIFY and re-quantified including the primary BM sample using VarScan (version 2.4.4 mpileup 2 snp and mpileup 2 indel), respectively. For detection of copy number and LOH events in primary and PDX samples, the R package Sequenza 4 was used. For subsequent quantification of these aberrations, averaged B allele frequency of heterozygous single-nucleotide polymorphisms (SNPs) was used. On average, a mean coverage of 80-fold was achieved for patient and PDX samples. For tracking of clonal evolution, individual clustering of VAFs for consecutive samples of each PDX and the corresponding transplanted patient sample was performed using the R (version 3.6.3) SciClone package (version 1.1)[40].

### Myeloid panel sequencing

DNA was isolated using the AllPrep DNA/RNA mini or micro Kit or QIAamp DNA Micro Kit (Qiagen, 56304). For panel sequencing, 250 ng of genomic DNA from patient and xenograft samples was subjected to the Nextera DNA Flex Kit with the usage of unique dual indices. The enrichment was performed using the IDT Hybridization Capture protocol and a corresponding custom myeloid panel (IDT, Integrated DNA Technologies, Coralville, IA, USA) including the following genes: *ASXL1, ASXL2, ATRX, BCOR, BCORL1, BRAF, BRCC3, CALR, CBL, CDH23, CDKN2A, CEBPA, CREBBP, CSF3R, CSNK1A1, CTCF, CUX1, DDX41, DDX54, DHX29, DNMT3A, EP3OO, ETNK1, ETV6, EZH2, FLT3, GATA1, GATA2, GNAS, GNB1, IDH1, IDH2, JAK2, KDM5A, KDM6A, KIT, KMT2D, KRAS, MPL, MYC, NF1, NPM1, NRAS, PHF6, PIGA, PPM1D, PRPF8, PTPN11, RAD21, RUNX1, SETBP1, SF1, SF3A1, SF3B1, SH2B3, SMC1A, SMC3, SRSF2, STAG2, SUZ12, TET2, TP53, U2AF1, U2AF2, WT1, ZBTB7A, ZRSR2*. The final library pools were sequenced on a S4 Nova Seq Flow Cell (Illumina) with 150 bp paired end reads. Mean sequencing depth was 1532.9-fold. Bioinformatical processing consisted of quality trimming using Seqtk (version 1.2) and was followed by a comprehensive quality control by using the FastQC package (version 0.11.5). Known false positive variants or single nucleotide polymorphisms were filtered out.

### Gomori silver impregnation staining

Deparaffinized sections were stained for 5 min in 0.5% potassium permanganate solution, 5 min in 2% oxalic acid solution, and 5 min in 5% ammonium iron sulfate solution. The samples were washed for 5 s in 10% ammoniacal silver nitrate solution and rinsed in distilled water. The slides were incubated for 5 min with 5% buffered formalin and for 2 min in 5% sodium thiosulfate. Between all incubation steps, the samples were rinsed with distilled water. The sections were counterstained for 10 min with 0.1% nuclear fast red-aluminum sulfate solution. Images were acquired using a PreciPoint M8 microscope and scanner. The images were quantitatively evaluated using QuPath software (version 0.3.0, https://qupath.github.io/). The whole section was covered with a grid with the tile size of 40 pixels, and the

percentage of squares (tiles) containing the areas of fibrosis or intra marrow ossifications were calculated by the operator blinded to treatment groups.

### Human mitochondria staining

Deparaffinized sections were stained using the anti-human mitochondria antibody (Clone 113-1; Sigma-Aldrich, MAB1273, 1:80) and detected with the EnVision Detection Systems Peroxidase/DAB, Rabbit/Mouse (Agilent). Tissues were counterstained with hematoxylin. Images were acquired using a PreciPoint M8 microscope and scanner. The intensity of mitochondrial staining in each section was analyzed using Aperio ImageScope software (Leica Biosystems, version 12.4.3.5008) and a positive pixel count algorithm with a color saturation threshold of 0.4.

### ECM preparation and immunofluorescent (IF) staining

For ECM preparation, MSCs were seeded in 24-well plates or 100 mm culture dishes at a concentration of $10^4$ cells/cm$^2$ an cultured for 14 days with or without daily PXS-5505 treatment. Confluent monolayers were rinsed 1X with PBS, decellularized using 20 mM ammonia for 20 min, RT and gently rinsed 2X with PBS. ECM was treated with 50U/ml DNase I (Roche Life Science, 4536282001) for 40 min, 37 °C and gently rinsed 2X with PBS. ECM images were acquired by Leica DMi1 inverted light microscope (Leica Microsystems) equipped with Leica HI PLAN I 20x/0.30 objective.

Sirius red staining of decellularized ECM was performed as described previously[41]. In details, ECM samples were fixed in 71% saturated picric acid solution, 8.5% formalin and 4.8% glacial acetic acid. Samples were washed with tap water (15 min) and stained in 0.1% Direct Red 80 reagent (Sigma-Aldrich, 365548) dissolved in saturated picric acid for 1 h, RT. The staining solution was discarded, and samples were washed extensively in 0.01 N hydrochloric acid.

For IF staining, ECM was fixed in 4% paraformaldehyde for 10 min, RT and blocked with 3% BSA for 60 min, RT. Fibronectin Alexa Fluor 488 antibody (Clone FN-3, ThermoFisher Scientific, 53-9869-80, 10 µg/ml) or collagen I polyclonal antibody (Sigma-Aldrich, AB745, 1:40) were diluted in PBS with 1% BSA and incubated overnight at 4 °C. Secondary goat anti-rabbit Alexa Fluor 488 IgG (H + L) polyclonal antibody (ThermoFisher Scientific, A-11034, 5 µg/ml in PBS with 1% BSA) were added for 2 h, RT. Images were obtained using Leica DMi8 fluorescence microscope (Leica Microsystems) equipped with Leica HI PLAN I 20x/0.30 objective. Images were acquired using Leica LAS X 3.4.1.17822 software.

### BM CD34 + HSPC culture on ECM and inhibition assays

BM CD34 + HSPCs were seeded on decellularized ECM derived from PXS-5505 or vehicle (PBS) treated MSCs in 24-well plates ($1×10^4$ cells/well) and cultured for 4 days in the presence of 5-AZA, PXS-5505 or P + A. In some of the assays, the following inhibitors and blocking antibody were used: myosin II inhibitor (±)-Blebbistatin (Sigma-Aldrich, 203390, 2 µM daily), ROCK inhibitor Y-27632 (Sigma-Aldrich, Y0503, 5 µM every 48 h), anti-integrin αVβ3 antibody (Clone LM609, azide-free, Sigma-Aldrich, MAB1976Z, 17 µg/ml as a single treatment) or mouse IgG1-k negative control (Clone MOPC-21, azide free, Sigma-Aldrich, MABF1081Z, 17 µg/ml as a single treatment). All inhibitors and antibody were added in the beginning of CD34 + HSPC/ECM co-cultures. After 4 days, CD34 + HSPCs were gently removed from ECM by pipetting and used for the CFU assay analyzed by flow cytometry.

### In-Gel digestion and mass spectrometry (MS)

ECM samples were harvested using XT sample buffer 4x (Bio-Rad, 161-0791) and heated to 95 °C for 5 min, cooled on ice and resolved on 4-12% gels using sodium dodecyl-sulfate polyacrylamide gel electrophoresis. Proteins were fixed within the gels using 5% acetic acid in 1:1 (volume/volume [v/v]) water:methanol for 30 min. After

Coomassie staining for 60 min, the gel slab was rinsed with water for 60 min, and the lanes were excised and cut into small pieces.

Subsequently, the proteins were in-gel destained using 100 mM ammonium bicarbonate/acetonitrile 1:1 (v/v), reduced in 10 mM dithiothreitol, alkylated using 50 mM iodoacetamide and finally trypsin digested by overnight incubation at 37 °C. The generated peptides were further subjected to a peptide extraction step with an acidic formic acid (1.5%) and acetonitrile (66%) solution. Peptide containing samples were dried in a vacuum centrifuge.

Dried peptides were re-dissolved in 0.1% trifluoroacetic acid and loaded on a C18 column (Phenomenex, Kinetex XB-C18, 150×0.3 mm) by direct injection using an Eksigent Ekspert NanoLC 425 system (AB Sciex). Peptides were eluted with an aqueous-organic gradient (4–48% acetonitrile in 0.1% formic acid, 125 min), at a flow rate of 5 μl/min and electrosprayed into a TripleTOF 6600+ mass spectrometer (AB Sciex). Each scan cycle consisted of one TOF-MS full scan and up to ten product ion dependent (IDA) MS/MS scans of the most intense ions. The mass spectrometer was run in the high sensitivity mode and the dynamic exclusion was set to 15 s. All analyses were performed in positive ion mode.

To generate the ion library, extracted MS/MS spectra were screened against the reviewed Uniprot Human database using the ProteinPilot search engine (AB Sciex, version 5.0.2) accepting cysteine alkylation and common biological modifications. All protein identification experiments were carried out using the corresponding decoy database and a false discovery rate (FDR) of 1%.

The SWATH acquisition was performed for an m/z range 400–1250 Da using looped isolation windows of 20 Da. The acquired data were processed with the SWATH Acquisition MicroApp 2.0 in PeakView Software (AB Sciex, version 2.2.0) using a spectral ion library generated from prior data-dependent acquisitions. Protein identification in SWATH was based on the following parameters: 4 peptides per protein, 3 transitions per peptide, 99% peptide confidence, 1% FDR, fragment ion extraction window of 5 min, and mass tolerance of 50 ppm.

Protein ion intensity data were imported into MarkerView (AB Sciex, version 1.3.1) to perform most likely ratio (MLR) normalization and group differences were examined using standard t-test. MS signal intensity values for each detected protein were normalized to the total MS signal in each sample.

### Scanning electronic microscopy (SEM)
The MSCs were seeded on the coverslips (SARSTEDT, 83.1840) and cultured for 2 weeks with or without daily PXS-5505 treatment. After decellularization of MSCs, ECM was washed once with PBS (1X). Samples were immersed in glutaraldehyde solution (Sigma-Aldrich, G5882; 2.5 wt. %, 1 ml) for 60 min, 500 μl/well. Samples were washed once with 500 μl PBS (1X). Complete dehydration was performed with increasing concentrations of ethanol (30%, 50%, 70%, 85%, 95%, and 100%), 10 min for each step, 500 μl/well, followed by hexamethyldisilazane (HMDS, Sigma-Aldrich, 379212; 50% HMDS in ethanol and 100% HMDS, 10 min each). HDMS solution was aspirated and samples were completely dried before analysis. The morphology of dehydrated ECM was characterized by a scanning electron microscope (Zeiss LEO 1530) at an operating voltage of 2 kV. Prior to the SEM measurements, the samples were coated with a 10 nm thick platinum layer.

### Statistical analysis
Statistical analysis was performed using GraphPad Prism 8.4.3 software (San Diego, CA, USA). Statistical comparisons of continuous variables between two groups were performed using two-sided Student's t test for unpaired parametric data or two-sided Mann-Whitney U test for unpaired non-parametric data. For comparison of continuous variables between multiple groups, the following tests were used: repeated measure (RM) one-way ANOVA with Tukey's multiple comparisons for matched parametric data, ordinary one-way ANOVA with Tukey's multiple comparisons for unmatched parametric data, Kruskal-Wallis test with Dunn's multiple comparisons for unmatched non-parametric data and Friedman test with Dunn's multiple comparisons for matched non-parametric data. Fisher's exact test (GraphPad Prism) was used for categorical variables to get RR with 95% CI. RR < 1.00 indicated a better therapeutic response. The type of statistical test is specified in figure legends. $P < 0.05$ indicates statistical significance.

### Reporting summary
Further information on research design is available in the Nature Portfolio Reporting Summary linked to this article.

## Data availability
The whole exome-sequencing and myeloid panel sequencing data have been deposited in the EGA archive (Accession code: EGAS00001006174). These data are available according to the policy of the EGA platform. Access can be requested via EGA, but also by direct personal communication with the corresponding authors of the manuscript. Responses can be expected within 72 h. Mass spectrometry proteomics data have been deposited to the ProteomeXchange Consortium via the PRIDE partner repository (Accession code: PXD031217). Complete digital scans of mouse femurs as a source data for Fig. 4g are available at BioStudies EMBL-EBI portal (Accession code: S-BSST1021). The remaining data are available within the Article, Supplementary Information. Source data are provided with this paper.

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

## Acknowledgements

This work was supported by the funding from the H.W. & J. Hector Foundation (Weinheim) (Project M83) (D.N.), the Deutsche Forschungsgemeinschaft (DFG) (No. 817/5-2, FOR2033, NICHEM) (D.N.), the "Forum Gesundheitsstandort Baden-Württemberg, Projektvorhaben „Identifizierung und Nutzung molekularer und biologischer Muster für die individuelle Krebsbehandlung" BW 4-5400/136/1 (D.N.), the German cancer aid foundation (Deutsche Krebshilfe, 70113953) (D.N.), the Gutermuth Foundation (D.N.), the Dr. Rolf M. Schwiete Foundation (Mannheim) (D.N.) and the Wilhelm Sander Foundation (2020.089.1) (J.C.J.). This work was supported by the Health + Life Science Alliance Heidelberg Mannheim and received state funds approved by the State Parliament of Baden-Württemberg (V.R.). D.N. is an endowed Professor of the German José-Carreras-Foundation (DJCLSH03/01). P.W. received research funding by the German Red Cross Blood Service Baden-Württemberg – Hessen. Q.X. was supported by the China Scholarship Council. P.A.L. thanks DFG (Heisenbergprofessur; 406232485, LE 2936/9-1) for the financial support. M.K. thanks the Excellence Cluster "3D Matter Made to Order" (2082/1-390761711) and the Carl Zeiss Foundation for financial support. The authors gratefully acknowledge the support of Stefanie Uhlig, operating the Mannheimer FlowCore Facility and the group of Prof. Adelheid Cerwenka for sharing the FACSAria™ Fusion Cell Sorter. For the publication fee we acknowledge financial support by Deutsche Forschungsgemeinschaft within the funding programme "Open Access Publikationskosten" as well as by Heidelberg University.

## Author contributions

Q.X., V.R., and D.N. designed and conducted the study, analyzed the data and wrote manuscript draft; A.S. performed bioinformatics analysis and wrote manuscript draft; J.C.J. performed bioinformatic analyses; E.A. and J.F. designed the study and contributed to experimental design; N.S. performed data analyses; C.S.A., F.R., J.W., and E.S. contributed to the study design; V.N., N.W., J.O., and I.P. provided technical assistance for sample workup, cell culture, and molecular analyses; M.K. and P.A.L. performed and analyzed the SEM experiments; A.J. and A.D. provided primary material from healthy controls; C.A.W. and A.M. provided histology and immunohistochemistry resources; V.C., E.J., and M.N. provided facility and technical support for mass spectrometry analysis; A.F. provided expertise in the evaluation of cytogenetics data; G.M., F.N., L.S., M.J., and P.W. provided material from patients or voluntary donors and clinical data; V.R. designed and supervised the study, performed experiments, analyzed the data and wrote manuscript draft; D.N. and W.K.H. supervised the study and provided research infrastructure.

## Funding

## Competing interests

This study was partly funded by research support of Pharmaxis Ltd., Australia. DN has stock ownership in Pharmaxis Ltd. The remaining authors declare no other competing interests.
