## [Peer Review File · Nature Communications]

Inhibition of lysyl oxidases synergizes with 5-azacytidine to restore erythropoiesis in myelodysplastic and myeloid malignanciesEditorial Note: Figures 1 and 2 from this document have been redacted at author's request.

REVIEWER COMMENTS

Reviewer #1 (Remarks to the Author); expert in bone marrow microenvironment and therapy:

This is a potentially interesting manuscript on role of Lysyl oxidases (LOX and LOX-like proteins which catalyse collagen crosslinking) in restoring erythropoiesis in myeloid proliferative neoplasms (MPN), especially MDS and PCV. Overexpression of LOX/LOXL genes have previously been reported in primary myelofibrosis (PMF) mediating cross-linking of ECM and contributing to fibrosis. Several studies have investigated blocking of LOX/LOXL to improve fibrotic outcomes (eg. current Phase ½ trials NCT04676529 with Pharmaxis antagonists and review PMID 33738462). The current manuscript appears unique in exploring whether (a) LOX/LOXL inhibitors may synergise with 5-AZA as an improved treatment strategy and (b) whether this combination may boost erythrocyte production as improved therapy for anaemic MN.

The manuscript contains 8 figures. At present some of these figures are not easy to interpret. Particularly presentation of Figures 5-8 could be revised to improve accessibility for readers.

Overall comments.

Fig 1. Show cultured MSC express LOX/LOXL mRNA and increased LOX/LOXL activity in BM plasma from MN patients.

Fig 2. Cell culture studies using 26 different MN patient CD34+ incubated in contact with (?autologous) MSC for impact of 5-AZA ± PXS-5505 on erythroid and megakaryocyte differentiation in vitro (using CFC-E, CFU-Meg and flow cytometry). Figure 2E-G shows 2A-C data as heat map and includes data from healthy normal patient cells (HY) as in vitro differentiation controls.

Fig 3. Similar in vitro cultures to Fig 2 performed, in order to determine whether the expanded erythroid cells had lower mutational load (represented less mutated clones) compared to the bulk HSPC readout from each patient.

Fig 4. Humanised mouse studies using patient HSPCs expanded on MSC in vitro then xeno-transplanted into NSG mice for impact of 5-AZA ± PXS-5505 treatment and on human HSPC engraftment. This is the most important experiment and figure of the manuscript (providing data for Figure 5 as well). A total of 63 NSG mice were engrafted with cultured HSPC+MSC from n=6 MN patients. No mice were engrafted with healthy normal "HY" donors in parallel. Inclusion of healthy controls would have strengthened the manuscript and conclusions. In this study it is not also clear why the prior step of HSPC+MSC co-culture was required prior to cell xeno-transplant and subsequent 5-AZA± PXS-5505 therapy in NSG mice (Fig 4A-G). An extra sentence explaining this rationale would help.

Readouts shown in Fig 4 include;

Fig 4C-D, interesting data showing reduced spleen size and slightly more red in femurs collected from PXS-5505 treated mice compared to untreated transplant controls. These data are encouraging re potential efficacy and suggest PXS-5505 treatment may reduce overall splenic enlargement and boost some erythropoiesis. Despit this it is not clear whether these readouts refer to the impact the injected agents have on normal mouse haematopoeisis compared in parallel with the engrafted human cells.

Fig 4E, % variant allele frequency (VAF); Fig 4G shows splenic engraftment (as correlation between '% splenic index' and '% engraftment' for each NSG mouse). Engraftment appears to correlate with splenic index. Though not clearly explained what '% splenic index' means. The coloured dots represent treatment groups, however on my copy the differences between the two red/pink groups (untreated, or P+A treated mice) were hard to distinguish, as where between the two blue groups. However one of the red/pink lines appear to engraft spleen better (? Untreated mice) than the others.

One of the blue lines was close to that (PXS-5505 group) suggesting this treatment does not affect engraftment, while AZA treatment alone or AZA + PXS-5505 treatment in mice resulted in lower engraftment. Stats (spearman rank) are shown but not clear whether they refer to differences in engraftment between the groups. As no transplant data from HPSC+MSC from normal "HY" donors was performed in NSG mice in parallel, it is hard to determine whether the PXS-5505 treatment in vivo may have any effect in transplant engraftment.

All mice were euthanised at 12 weeks. No data on potential impact of treatments on mouse morbidity, weight changes etc are shown. If these pre-clinical studies are to 'lay out a strong pre-clinical rationale for efficacy of combination treatment' as mentioned in last sentence of abstract it would be important that more data on overall impacts of the suggested treatment strategy on overall humanised NSG mouse wellbeing and overall disease burden and treatment responses. These studies could compare the treatments in both humanised mice, and MPN mouse-mouse studies (many models are available as a standard in field) plus the impacts of these agents on healthy HSPC as well as HPSC from MPN. The absence of these data diminishes the potential importance and impact of this overall manuscript.

Fig 5. Shows selected data from the humanised NSG mouse transplant study in Fig 4 referring to NSG transplants from patients 11, 3 and 30. Fig 5A is data direct from patient 11 humanised mice and confirms treatment of P11 transplanted NSG with 'F+A' combination leads to a boost in the proportion of erythroid (CD235a+, CD45-) cells compared to total non-differentiated human CD34+ CD45+ cells at euthanasia. AZA treatment alone slightly boosted proportion of human glycophorinA (CD235a+, CD45-) cells but PXS-5505 alone treatment did not. Figure 5 B, C, D are similar but readout taken following the extra step of ex vivo EPO expanded culture (from NSG engrafted with patient 11, 3, 30).

These Fig 5 data appear somewhat at odds with the femur and spleen colour differences suggested from (combined data from all PDX mice from patient donors) in Figure 4 and suggest that the total changes in engraftment between treatment groups in Fig 4 varies enormously are not necessarily associated with an increased proportion of human erythroid production. Could authors discuss these points. Could the authors specifically can the authors show spleen weights and femur 'colour' pictures from NSGs from patients 11,3 and 20 for each of the 4 different treatment groups in supplementary figures. This would help gain a feel for the overall variability of these data between donors and especially give more a feeling on whether the suggested treatment strategy may only benefit a few patients.

Figure 6. In vitro assays based on patient HSPC with MSC co-cultures followed by % erythropoiesis cell readouts. Question asked was whether the treatment altered erythroid differentiation via impacting patient MSC alone and whether all patient MSC responded the same with readouts being % erythroid differentiation on the co-cultured HSPC (readout shown as %CD235a+, CD45- cells).

The current figure is very hard and frustrating for this reviewer to follow. The key difficulty is the X-axis doesn't simply show what patient the HSPC cells were from (this should be listed on top row of X-axis) and what patient the MSC were from (should be listed on the row below the HSPC) on X-axis instead. This simple change would make the whole figure interpretable. Then it would be evident which HSPC were cultured with autologous (same patient number) or allogeneic (different MSC patient number).

Currently the figure is grouped into A, B, C based on whether MSC appear to respond to P+A treatment in vivo (ie. increase CD235a production in HSC). However it may be simpler to follow in the experiments from each HSPC were grouped together (meaning internal controls included at same location). It also may help with interpretation.

Eg. In Fig 6 it seems that CD34+ HSPC from Patient 3 (P3) responded well to; F+A treatment when co-cultured with auto (P3) MSC (shown in Fig 6A left column) and somewhat responding to (P25) MSC (shown in Fig 6C near left column), however did not respond to (P31) MSC after F+A treatment. Is this correct? Does this mean (P31) MSC are always non-responders to F+A?

Likewise is it correct that HSPC from patient (P11) ; do not respond to auto (P11) MSC after F+A, but do respond to treated (P25) MSC (Fig 6B left col) and somewhat respond to (P31) MSC (FIG 6A right column)? Overall suggesting that the MSC from this patient do not respond to F+A treatment?

Could the authors also please give their criteria for choosing each of the HPSC and MSC groups represented. Were these solely selected on availability of enough cells or selected more on whether the data fit with hypothesis. Similar comment for Figure 6.

Figure 7. examines impact of P+A treatment on ECM stiffness, collagen (Fig 6 A-E) and impact on erythroid % change in alone MSC, PXS-5505 treated MSC or with HSC co-culture HSPC for P3, P10 and P11 MSC after decellularisation (possibly data from same P3, P10, P11 experiment as shown in Fig 5) with readout being CFU-E assay instead in Fig 7F-J.

The colony assay readouts only include % CD235a+ 45- cells weight no reference to total number of colonies produced or HSPC survival. It would be ideal to show all readouts including total CFU numbers for each group for each treatment.

As several controls may not be included (only groups shown HSC separated by transwells it is hard to fully interpret data. Each treatment group could have matched untreated control.

To aid appreciation of the data it would be helpful if Fig 7 F-H were changed so the top tier contains data only from HSC+MSC co-cultured cells (including all treatment comparisons) then the bottom tier of figures be the matched HSC + decellularized ECM datas. This would help enormously with interpretation. At present the data seems to suggest that CD34+ cultured alone show superior differentiation to CD235a+45- cells (%) than PXS-5505 treated alone ECM. Is this correct interpretation (maybe easier once figure if re-organised) and please explain if so.

Figure 8 is mechanistic studies using blocking antibodies to proposed integrin avb3 interaction and signalling pathways. Only impact on % CD235a, CD45- cells shown. No data on overall HPSC survival in each of these cultures is shown, nor impacts of treatments on HSPC differentiation to other cell populations. integrin avb3 is not a completely novel target. Ideally blocking this integrin in mouse studies in vivo would be gold standard.

Reviewer #2 (Remarks to the Author); expert in MPN:

This is an interesting and thorough study by Xu et al. that addresses the mechanism by which combination treatment with lysyl oxidase and 5-AZA treatment functions in myeloid neoplasms. Overall this study is novel and timely, and the dissection of the cellular and molecular requirements for this combination treatment makes this work even stronger. I have several comments that I believe could strengthen the manuscript:

1. Figure 1: I understand the authors are investigating myeloid neoplasms as a whole here, but I think it would be helpful to see LOXL2 expression, concentration, and activity separated by disease subtype. For instance, in figure 1A, the error bar is quite big for the LOXL2 gene expression in MSCs. Can the authors show LOXL2 mRNA expression in MSCs for MDS versus CMML etc? Also, if possible, the authors should add more samples here (3-5 more would suffice) to strengthen the conclusion that LOXL2 is up-regulated in MSCs from MN patients.

2. Figures 1H and I: the authors should include healthy donors in these viability curves, and compare the response of patient cells to healthy donors. Additionally, the authors should include at least 2 patients per type of disease.

3. Figure 2A: The authors show in Figure 1 that PXS treatment leads to decreased collagen crosslinking in MDS patients with bone marrow fibrosis. In the co-culture shown in 2A, the authors should stain MSCs for reticulin (or even collagen expression) to determine whether PXS decreases collagen deposition in bone marrow stromal cells.

4. Figure 4: in this mouse model, do mice develop bone marrow fibrosis? If so, does this combo treatment prevent or reverse fibrosis in addition to its effects on erythroid differentiation? If so, the

authors should highlight this finding. If this mouse model does develop BM fibrosis but the combo treatment doesn't affect it, the authors should comment on this.

5. Figure 7: If PXS treatment increases collagen I expression in MCSc, how do the authors reconcile this with their claim that PXS treatment reduces MN disease burden? Increased MSC collagen expression seems to me an indication of increased MN disease burden. The authors should comment on this in the discussion.

Reviewer #3 (Remarks to the Author); expert in extracellular matrix:

This is a wide ranging and detailed pre-clinical study using patient-derived bone marrow cells to investigate synergy between azacytidine and the pan-lysyl oxidase inhibitor PXS-5505 to ultimately potentially improve clinical outcomes in myelodysplastic diseases. The manuscript is clearly written, though this reviewer would have preferred fewer abbreviations. The use of PXS-5505 instead of the typical BAPN is a strength in light of the relative toxicity profiles of the two inhibitors, and potential clinical translation. The comprehensive nature of the assays performed in vitro and in xenografts is to be commended, as is transparency of data presentation. Below are some comments/questions for the authors to consider in a (most likely) minor revision.

1. Why were only 4 healthy control subjects enrolled in this study? Does this provide sufficient data to make firm conclusions?
2. Was LOX family enzyme activity determined in Azacytidine-treated only MN bone marrow plasma cells compared to non-treated bone marrow plasma cells? Azacytidine is known to up-regulate LOX family members in some, but not all tumor cells. If Azacytidine upregulation of LOXs occurs in MN, an explanation regarding why the combination of azacytidine + PXS-5505 results in a better outcome could be that PXS-5505 may prevent lysyl oxidase-driven fibrosis that is otherwise stimulated by Azacytidine.
3. Figure 1C. Did lysyl oxidase enzyme activity assays include incubations +/- BAPN?

General comments to all reviewers from the authors:

We thank you very much for taking the time to review our manuscript and for making many very constructive suggestions for improvement. Before we reply point by point to all suggestions, we would like to make you aware of two changes to the manuscript, which we suggest pro-actively independently of your suggestions for improvement:

1.

We would like to suggest to adapt the title of the manuscript by addition of the term “myelodysplastic” neoplasms. We suggest this amendment in acknowledgement of the updated classification systems published in parallel to the revision of this manuscript:

The 5th edition of the World Health Organization Classification of Haematolymphoid Tumours: Myeloid and Histiocytic/Dendritic Neoplasms, *Leukemia* (2022) 36:1703–1719; <https://doi.org/10.1038/s41375-022-01613-1>

International Consensus Classification of Myeloid Neoplasms and Acute Leukemias. *Blood*. 2022 Sep 15;140(11):1200-1228. doi: 10.1182/blood.2022015850.

Before the publication of these novel classifications we described the portfolio of diseases in our study (MDS, MDS/MPN, CMML, AML-MRC) as “myeloid neoplasms”, to acknowledge the fact that not all samples were Myelodysplastic Syndromes”. However, especially the new WHO classification now introduces the novel term “Myelodysplastic Neoplasms...”, which share common biology: quote: “...to replace myelodysplastic syndromes, underscoring their neoplastic nature and harmonizing terminology with MPN.” On the background of the high enrichment of MDS samples, AML with MRC and MDS/MPN overlap syndromes / CMML in our study, which are now all subsumed under the term “Myelodysplastic Neoplasms”, we believe that it is warranted to adapt the title of the manuscript to the new situation created by these updated disease classifications.

2.

Even though it wasn't specifically asked for, we also included novel electron microscopy images of healthy and MDS derived ECM +/- PXS-5505 treatment (*Results*: page 12; new supplementary Figure 17; new *Supplementary Methods*: pages 7-8), which we believe further improves the overall data. Therefore, we have included two additional co-authors in the authors list. All authors agree with these changes in the author list.

Point by point responses:

REVIEWER

Reviewer #1 (Remarks to the Author); expert in bone marrow microenvironment and therapy:

This is a potentially interesting manuscript on role of Lysyl oxidases (LOX and LOX-like proteins which catalyse collagen crosslinking) in restoring erythropoiesis in myeloid proliferative neoplasms (MPN), especially MDS and PCV. Overexpression of LOX/LOXL genes have previously been reported in primary myelofibrosis (PMF) mediating cross-linking of ECM and contributing to fibrosis.

Several studies have investigated blocking of LOX/LOXL to improve fibrotic outcomes (eg. current Phase ½ trials NCT04676529 with Pharmaxis antagonists and review PMID 33738462). The current manuscript appears unique in exploring whether (a) LOX/LOXL inhibitors may synergise with 5-AZA as an improved treatment strategy and (b) whether this combination may boost erythrocyte production as improved therapy for anaemic MN.

The manuscript contains 8 figures. At present some of these figures are not easy to interpret. Particularly presentation of Figures 5-8 could be revised to improve accessibility for readers.

Authors' Answer: We are thankful for the reviewer's helpful and constructive suggestions for improvement. Within the scope of this revision, we have carefully evaluated all comments of this reviewer and made amendments in Figures 5-8 in order to improve accessibility. The detail of amendments are outlined in detail below in response to all comments.

Reviewer #1:

Overall comments.

Fig 1. Show cultured MSC express LOX/LOXL mRNA and increased LOX/LOXL activity in BM plasma from MN patients.

Fig 2. Cell culture studies using 26 different MN patient CD34+ incubated in contact with (?autologous) MSC for impact of 5-AZA ± PXS-5505 on erythroid and megakaryocyte differentiation in vitro (using CFC-E, CFU-Meg and flow cytometry). Figure 2E-G shows 2A-C data as heat map and includes data from healthy normal patient cells (HY) as in vitro differentiation controls.

Authors' Answers: We would like to respectfully point out that we analyzed n=31 MN patients, some of which were analyzed at two different time points adding up to a total of n=33 data sets in Figure 2e, not only n=26.

With regard to the comment / question "(?autologous) ", we can answer that all standard co-incubation experiments were carried out with autologous MSCs unless otherwise specified. We apologize that this did not become clear. We found that we had not directly specified this in the corresponding results section for Figure 2. For increased clarity we now also specified it there to improve comprehensibility (Results: page 6, line 124; Methods: page 17, line 420; The legend of Figure 2: page 30).

Reviewer #1:

Fig 3. Similar in vitro cultures to Fig 2 performed, in order to determine whether the expanded erythroid cells had lower mutational load (represented less mutated clones) compared to the bulk HSPC readout from each patient.

Fig 4. Humanised mouse studies using patient HSPCs expanded on MSC in vitro then xeno-transplanted into NSG mice for impact of 5-AZA ± PXS-5505 treatment and on human HSPC engraftment. This is the most important experiment and figure of the manuscript (providing data for Figure 5 as well). A total of 63 NSG mice were engrafted with cultured HSPC+MSC from n=6 MN patients. No mice were engrafted with healthy normal "HY" donors in parallel. Inclusion of healthy controls would have strengthened the manuscript and conclusions. In this study it is not also clear why the prior step of HSPC+MSC co-culture was required prior to cell xeno-transplant and subsequent 5-AZA± PXS-5505 therapy in NSG mice (Fig 4A-G). An extra sentence explaining this rationale would help.

Authors' Answers: We thank the reviewer for this comment and acknowledgement of the importance of this pre-clinical in-vivo experiment. However, we would first like to clarify that we did not perform co-culture before xenotransplantation as outlined by this reviewer and apologize that this did not seem to be clear from the manuscript. Original non-expanded primary patient derived HSPCs were used for xenotransplantation. Only MSCs were in vitro expanded but not pre-treated before injection into NSG mice. As amending action we have now further clarified this in new Figure 4a, and in the Methods section (page 16, subtitle "*Xenotransplantation experiments*").

As a further point of criticism, this reviewer here raises an absence of healthy xenograft transplants as an issue that weakens the manuscript and conclusions. However, the necessity and reason for such controls was not explained. We therefore assume that this comment may be aimed at ruling out potentially unknown toxicity issues especially of our novel proposed drug combination on healthy hematopoiesis. If this interpretation is correct, we agree that this issue is of utmost importance and have therefore taken multiple steps, in order to more thoroughly address the question of potential hematologic toxicity of the propagated substances and their combination PXS-5505 + 5-AZA by the following measures:

1.

We have added further n=3 primary human healthy donor samples (HY5-HY7) to the in-vitro data in new Table 1, new Figure 2e, h, as well as new Supplementary Figure 4d, 5 and 7d, which display hematopoietic differentiation of healthy human HSPCs in autologous MSC/HSPC co-cultures and show that the differentiation capacity and hematopoietic output is neither perturbed by the single substances nor the combination PXS-5505 + 5-AZA.

2.

We have generated a completely new Supplementary Figure 12 and 13 (*Results*: page 10, line 225-231) presenting comprehensive clinical data / wellbeing data of the xenograft-bearing mice in our PDX experiments. Besides routine blood counts, we have performed additional flow-cytometric measurements from archived blood and bone marrow samples to analyze endogenous murine hematopoiesis. In summary, these results clearly showed that animal well being as well as hematopoiesis remained stable throughout the experiments under all conditions. Specifically, our proposed substance combination PXS-5505 + 5-AZA was not more toxic than the single substances alone.

3.

We provide a completely novel data set of long-term (26 weeks) PXS-5505 administration in healthy rats, which demonstrates that PXS-5505 does not affect hematological parameters (new Supplementary Figure 14; *Results*: page 10, line 231-233; *Methods*: page 17, subtitle "*PXS-5505 toxicity studies in rats*").

4.

Ultimately, PXS-5505 has already been tested as a single substance in humans within the scope of early phase clinical trials and showed an excellent safety profile with no adverse effects on healthy human hematopoiesis. The report for this clinical study is publicly available at the Australian New Zealand Clinical Trials Registry (ACTRN12619000332123) and was cited in the Discussion section (page 15, line 367-371). We also enclosed it here as a separate file for the reviewer's evaluation (named as "Reviewer 1 only, ACTRN PXS trial on healthy volunteers_Basic results summary report"). The assessment of hematological parameters can be found on the pages 9-11 of this report.

5.

Besides safety studies in humans, the effects of PXS-5505 on mouse hematopoiesis have been evaluated in a previous study in a murine myelofibrosis model / JAK2V617F mice (Int J Hematol. 2019 Dec;110(6):699-708. doi: 10.1007/s12185-019-02751-6. Epub 2019 Oct 21;). PXS-5505 treatment alone did not affect blood counts, including white blood cell count (WBC), hemoglobin

(Hb), red blood cell counts (RBCs), hematocrit and platelet levels in both males and females. Additional text and reference was added in Discussion section, pages 15-16 (line 371-373) and References section, page 24.

In summary, we hope that by adding the data outlined above, we were able to settle concerns about a potentially increased toxicity issue of the proposed substance combination on healthy hematopoiesis.

A specific complete de novo repetition of xenograft experiments with healthy donor samples as suggested by this reviewer would take us a minimum of 9-12 months of additional experimental time. This is clearly beyond the scope of reasonable revisions and almost adds no relevant information to what we have already supplied. To this end, we also formally rebut, that absence of healthy xenografts weakens our manuscript: Apart from the additional data we now provided in response to this criticism, all experiments in our study were thoroughly controlled by “non-treatment” controls – allowing a direct comparison of the drug effects in vitro and in the in-vivo PDX experiments. Throughout our study in all experiments controlled with non-treatment controls we observed that instead of being toxic, our proposed substance combination synergistically improves hematopoiesis in the primary human patient samples as compared to non-treatment control. All experimental animals were clinically well under all treatment conditions. A reason to further assess the drug combination in healthy xenografts is therefore not clear to us because this would also not be part of clinical trial to test this substance combination in humans. The reason for this is that the toxicity profile of 5-AZA (being a cytotoxic drug) is well known since over 10 years of clinical application with its known side effects. PXS-5505 on the other hand, has already been tested in early clinical trials in humans and has a completely favorable safety profile. Therefore, also in the planned clinical trial based on the results of this manuscript, there will be no tests in healthy individuals due to the well-known and previously thoroughly characterized hematotoxicity profile of 5-AZA.

Reviewer #1:

Readouts shown in Fig 4 include;

Fig 4C-D, interesting data showing reduced spleen size and slightly more red in femurs collected from PXS-5505 treated mice compared to untreated transplant controls. These data are encouraging re potential efficacy and suggest PXS-5505 treatment may reduce overall splenic enlargement and boost some erythropoiesis. Despit this it is not clear whether these readouts refer to the impact the injected agents have on normal mouse haematopoeisis compared in parallel with the engrafted human cells.

Authors' Answer: We thank the reviewer for pointing out that the purpose of the readouts in Figures 4C-D did not become clear. We fully agree, that without further explanation and experimental analysis, the depiction of exemplary femurs was missing clarity. We therefore decided to remove the mouse femur images shown in the initially submitted Figure 4D because the more intense red bone color could not only be caused by boosted erythropoiesis, but also by lower engraftment rates in 5-AZA and P+A arms. Therefore, these images alone do not provide objective unambiguous support for the induction of human erythropoiesis.

As an improved replacement for the removed images we have performed elaborate additional experiments and revisions in order to improve data presentation and discriminate between effects on human and murine hematopoiesis resulting in new Supplementary Figure 13a-e and new Supplementary Figure 12 for flow cytometry gating strategies of mouse hematopoiesis in our PDX models (*Results*: page 10, line 225-231). There, we now provide comprehensive data on all parameters of the murine hematopoiesis in the performed xenografted studies. Our data show that

both PXS-5505 and P+A combination treatments did not affect murine hematopoiesis, including erythroid cells, myeloid cells, megakaryocytes, hemoglobin and platelets. Moreover, we further validated the specificity of analysis methods of the human hematopoiesis within the PDX studies. All of our antibodies were tested to have a very high specificity for human markers in assessment of the cross-reactivity using flow cytometry (new Supplementary Figure 11) and could well distinguish mouse and human hematopoietic cells and progenitors. Therefore, our experiments specifically detected the effect of P+A treatment on the restoration of a human MN-derived erythropoiesis.

In summary, the removal of the femur images and replacement by new data with increased specificity does not change our initial conclusions or messages in the manuscript. On the contrary they strengthen the other presented quantitative data, such as spleen indices, engraftment and mutational changes.

Reviewer #1:

Fig 4E, % variant allele frequency (VAF); Fig 4G shows splenic engraftment (as correlation between ' % splenic index' and ' % engraftment' for each NSG mouse). Engraftment appears to correlate with splenic index. Though not clearly explained what ' % splenic index' means. The coloured dots represent treatment groups, however on my copy the differences between the two red/pink groups (untreated, or P+A treated mice) were hard to distinguish, as where between the two blue groups. However one of the red/pink lines appear to engraft spleen better (? Untreated mice) than the others. One of the blue lines was close thought (PXS-5505 group) suggesting this treatment does not affect engraftment, while AZA treatment alone or AZA + PXS-5505 treatment in mice resulted in lower engraftment. Stats (spearman rank) are shown but not clear whether they refer to differences in engraftment between the groups. As no transplant data from HPSC+MSC from normal "HY" donors was performed in NSG mice in parallel, it is hard to determine whether the PXS-5505 treatment in vivo may have any effect in transplant engraftment.

Authors' Answer: We thank the reviewer for the detailed comments regarding Figure 4. After careful evaluation of this comment, we have decided to address the criticism of original Figure 4G by deleting this figure because we are in agreement that this figure causes more confusion than benefit for understanding the data.

For better understanding and also in response to a further comment by this reviewer below requesting depiction spleen indices for all patients and conditions individually, we have expanded the initial Figure 4H to new Figure 4f to include spleen indices from all treatment conditions and individual patients (*Results*: page 9, line 198-205).

For explanation: The original Figure 4G showed a correlation of splenic index with bone marrow engraftment (not human engraftment in spleen) at the endpoint after completion of treatment. Purpose of this figure was to demonstrate that bone marrow engraftment and spleen indices correlated well and that they could therefore be used concomitantly as a reflection of disease activity in-vivo. While it was defined in the figure legend what spleen index means (The legend of Figure 4: Page 31; *Results*: page 9, line 199-200; "Spleen index [%] = Spleen weight/body weight*100"), we fully agree that from the results description, this may have been difficult to understand.

The original Figure 4G was only a correlation of the two clinical parameters, and therefore it is not well suited for making comparisons with regard to the readouts in dependency of the different treatment conditions as outlined by the reviewer, i.e. it leads to confusions, when trying to read the original Figure 4G in terms of efficacy differences between the treatment conditions. This is only possible in original Figures 4F and 4H, in which a direct comparison of the different treatment conditions was performed on the single parameters.

To this end, original Figure 4F (now new Figure 4e) also answers the reviewers question of any effect of PXS-5505 treatment on transplant engraftment: Figure 4F showed that PXS-5505 did not have any effect on engraftment as compared to non-treatment control.

Reviewer #1:

All mice were euthanised at 12 weeks. No data on potential impact of treatments on mouse morbidity, weight changes etc are shown. If these pre-clinical studies are to 'lay out a strong pre-clinical rationale for efficacy of combination treatment' as mentioned in last sentence of abstract it would be important that more data on overall impacts of the suggested treatment strategy on overall humanised NSG mouse wellbeing and overall disease burden and treatment responses. These studies could compare the treatments in both humanised mice, and MPN mouse-mouse studies (many models are available as a standard in field) plus the impacts of these agents on healthy HSPC as well as HSPC from MPN. The absence of these data diminishes the potential importance and impact of this overall manuscript.

Authors' Answer: The comment that "...All mice were euthanised at 12 weeks..." does not reflect the fact that we performed long-term experiments. Mice were transplanted with patient samples at the age of 8-10 weeks, then transplants were left 12 weeks for establishment of robust long-term engraftment, then mice were treated for 12 weeks, which corresponds to 3 full cycles of 5-AZA +/- PXS-5505 corresponding to the real life clinical treatment protocols of MDS with 5-AZA. That means, mice were euthanized between week 32 – 34.

We fully agree that that mouse morbidity (weight changes etc.) is an important point. As already outlined above, we have addressed this point of criticism with elaborate additional experiments and data to evaluate the tolerability of our proposed substance combination on mouse well-being in these experiments:

1. Completely new Supplementary Figure 13 (*Results*: page 10, line 225-231) presenting comprehensive clinical data / wellbeing data of the xenograft-bearing mice in our PDX experiments.

Importantly, these data convincingly show that none of the treatment conditions, including the combination treatment arm, had negative effects on mouse health as evidenced by stable body weight and stable endogenous hematopoiesis.

2. Completely novel data set of long-term (26 weeks) PXS-5505 administration in healthy rats (new Supplementary Figure 14; *Results*: page 10, line 231-233).

3. Referral to Phase I clinical trial data on tolerability of PXS-5505 in humans (ACTRN12619000332123; *Discussion*: page 15, line 367-371; please also see the separate file for the reviewer's evaluation: "Reviewer 1 only, ACTRN PXS trial on healthy volunteers_Basic results summary report").

4. Addition of further healthy controls to the in-vitro experiments (new Table 1, new Figure 2e, h, as well as new Supplementary Figure 4d, 5 and 7d).

5. Addition of non-treatment controls in experiments, which were still missing these (new Supplementary Figure 16a).

Finally, we would like to reply that 5-AZA has been part of guideline treatment of MDS, CMML and AML of the elderly for over 10 years with a well characterized profile of known side effects. 5-AZA

is so far the only substance leading to a survival benefit in MDS patients. However, due to its property as a cytidine analog, 5-AZA is a cytotoxic agent with the side effects anemia, neutropenia and thrombocytopenia. Similarly to classical chemotherapy it will therefore never be applied to healthy individuals. Therefore, we disagree that the absence of an additional control group with healthy HSPCs in our xenograft experiments is a weakness in our study. Our study has been controlled by non-treatment controls in all experiments yielding all necessary information on effects on disease burden and treatment response exerted by the single substances and their combination.

With regard to reproduction of these experiments pure murine models, we disagree that absence of repetition of our experiments in a pure murine model is a weakness of our study. On the contrary, transgenic mouse models of MN are predominantly monogenic and therefore do not reflect the molecular heterogeneity of real life MDS / CMML and AML samples as used in our current study. Nevertheless, one of the most frequently used models is the NUP98/HOXD13 (NHD13) transgenic model of MDS and experiments with this model are currently underway in our lab with the predominant aim to provide more insights on mechanisms underlying the therapeutic effects of the P+A combination. This ongoing study aims to involve over 25 mice per treatment arm for robust statistical assessment of P+A effects on the mouse survival as well as MDS-derived hematopoiesis. However, it is not clear yet, whether this model will enable a reproduction of the results inferred in human samples at all because this model is a monogenic model and likewise artificial in comparison to human PDX / in-vitro models. The acquisition of sufficient data for survival analyses will take around two years because "MDS"-carrying mice usually succumb from the disease at the age of 10-12 months. Therefore, we plan to publish these results as a separate manuscript with a different (hopefully mechanistic) focus. However, we are happy to provide the reviewer with initial data from the first experiment in this model for "reviewer-only" evaluation. The current preliminary data for n=7-10 mice per treatment arm show beneficial effects of PXS-5505 on the time of recovery from 5-AZA associated toxicity (Reviewer-only Figure 1A below). The data also show the absence of PXS-5505 and P+A effects on mouse body weights (Reviewer-only Figure 1B below), indicating no toxicity of PXS-5505 treatment in NHD13 mice. There is also a tendency for superior survival in P+A treatment arm (Reviewer-only Figure 1C below), which requires higher sample size for adequate statistical power. Besides, the leukemia-free survival was comparable between the three treatment arms, indicating no effect of PXS-5505 or P+A on the morbidity of leukemia transition in NHD13 mice (Reviewer-only Figure 1D below). At the same time, we also assessed the effects of the treatments on blood parameters (Reviewer-only Figure 2A-C below) and hematopoiesis in the bone marrow (Reviewer-only Figure 2D below) in NHD13 mice. These data show potential beneficial effects of P+A on the occurrence of "remission" events in leukemic NHD13 mice (leukemia transition was indicated by high white blood cell counts $\geq 10 \times 10^6/\text{dl}$) as assessed by reduction of high WBC counts in 3 out of 4 P+A treated mice that transitioned into leukemia (Reviewer-only Figure 2C below). Moreover, the data suggest potential induction of erythropoiesis by P+A in these mice (Reviewer-only Figure 2D below). Due to the preliminary nature of this data, long time to completion and separate aim of the experiments, we hope that the reviewer understands, that these results cannot be included into the current study and that for competitive reasons, we cannot wait until these experiments will be concluded.

Figure 1 redacted at author's wishes

Figure 2 redacted at author's wishes

Reviewer #1:

Fig 5. Shows selected data from the humanised NSG mouse transplant study in Fig 4 referring to NSG transplants from patients 11, 3 and 30. Fig 5A is data direct from patient 11 humanised mice and confirms treatment of P11 transplanted NSG with 'F+A' combination lead to a boost in the proportion of erythroid (CD235a+, CD45-) cells compared to total non-differentiated human CD34+ CD45+ cells at euthanasia. AZA treatment alone slightly boosted proportion of human glycophorinA (CD235a+, CD45-) cells but PXS-5505 alone treatment did not. Figure 5 B, C, D are similar but readout taken following the extra step of ex vivo EPO expanded culture (from NSG engrafted with patient 11, 3, 30).

These Fig 5 data appear somewhat at odds with the femur and spleen colour differences suggested from (combined data from all PDX mice from patient donors) in Figure 4 and suggest that the total changes in engraftment between treatment groups in Fig 4 varies enormously are not necessarily associated with an increased proportion of human erythroid production. Could authors discuss these points. Could the authors specifically can the authors show spleen weights and femur 'colour' pictures from NSGs from patients 11,3 and 20 for each of the 4 different treatment groups in supplementary figures. This would help gain a feel for the overall variability of these data between donors and especially give more a feeling on whether the suggested treatment strategy may only benefit a few patients.

Authors' answer: As outlined above to the previous comment of this reviewer we decided to remove the initially submitted panel D showing femurs from original Figure 4 because it does not provide objective / unambiguous support for P+A effects on erythropoiesis. We agree with the reviewer regarding the heterogeneity of engraftment rates and engraftment changes in the course of treatment. However, this heterogeneity is expected and fully represents the biological heterogeneity of primary patient samples derived from MDS, CMML and AML. Patient derived xenografts from these diseases are known to be difficult to establish and heterogeneous, but nevertheless, these models are one of the strongest preclinical platforms for analyzing these diseases with high faithfulness in replication the clinical "real life" situation (Gbyli et al, doi: 10.1016/j.bcp.2020.113794; Lee and Deeg, DOI: 10.1016/j.exphem.2013.10.002, Belotserkovskaya, Demidov, DOI: 10.3390/ijms222111510 and others).

Moreover, this heterogeneity might be further augmented in mice due to additional reasons (e.g. possible difference in engraftment of certain HSPCs clones that contribute to the residual erythropoiesis as well as the difference in the percentage of erythroid colony- initiating human CD34+ HSPCs that survive in the bone marrow to the treatment endpoint). We also think that it is important to uncouple the effects of P+A treatment on the disease burden (engraftment rates, spleen sizes and indice) from its effects on the erythropoiesis. The reduction in the dominant mutated clone size by P+A with concomitant decrease in the engraftment rates leads to reduced disease burden. However, residual HSPC clones may not be capable of erythroid differentiation.

To accommodate the reviewers comments the following measures were taken:

1. During the revision experiments we additionally found increased human erythroid differentiation in xenografts of P32 treated with P+A. This data was presented in new Figure 5b and page 10 (line 220-222) of Results section.
2. Additional discussion of the observed heterogeneity was also added in the manuscript to the "Discussion" section, page 15 (line 357-359), as suggested by the reviewer.
3. Spleen indices were provided on an individual per patient level in new Figure 4f as requested by the reviewer.

With regard to the heterogeneity observed therein and fraction of patients benefitting from the P+A combination, we would like to point out that the main message of this manuscript (i.e. comparable to “primary endpoint”) is the increase of erythropoietic differentiation induced by the substance combination P+A. The NSG PDX model is not optimal to explore this parameter since human erythrocytes are eliminated by macrophages in this mouse model. This was clearly pointed out as a caveat in our original manuscript (current version of text, *Results*: page 9, line 218-219; Reference 25: Hu Z, Van Rooijen N, Yang Y-G. Macrophages prevent human red blood cell reconstitution in immunodeficient mice. *Blood* 118, 5938-5946 (2011)). To this end, original Figure 2E (new Figure 2e) actually answers the reviewers question much better: There, in addition to 29% of patients responding to 5-AZA alone, an additional 35% of patients had an erythroid response exclusively to the novel combination of P+A. This gives an estimation of which fraction of patients would possibly benefit from the new P+A combination. We would like to highlight that if such an effect size on erythroid improvement would even partially be reproduced in real life MDS patients, this would be a great achievement for MDS and comparable with recent large scale Phase III Studies, e.g. the MEDALIST Trial (doi: 10.1056/NEJMoa1908892.), which led to approval of Luspatercept for low risk MDS. In that trial, the primary endpoint (=8 weeks transfusion independence) was reached by 38% of patients with a background placebo response of 11%). This response rate is completely comparable with that of our current study with the addition that our substance combination achieved this in all MDS risk groups and in patient samples, which were partially heavily pre-treated.

Reviewer #1:

Figure 6. In vitro assays based on patient HSPC with MSC co-cultures followed by % erythropoiesis cell readouts. Question asked was whether the treatment altered erythroid differentiation via impacting patient MSC alone and whether all patient MSC responded the same with readouts being % erythroid differentiation on the co-cultured HSPC (readout shown as %CD235a+, CD45- cells).

The current figure is very hard and frustrating for this reviewer to follow. The key difficulty is the X-axis doesn't simply show what patient the HSPC cells were from (this should be listed on top row of X-axis) and what patient the MSC were from (should be listed on the row below the HSPC) on X-axis instead. This simple change would make the whole figure interpretable. Then it would be evident which HSPC were cultured with autologous (same patient number) or allogeneous (different MSC patient number).

Currently the figure is grouped into A, B, C based on whether MSC appear to respond to P+A treatment in vivo (ie. increase CD235a production in HSC). However it may be simpler to follow in the experiments from each HSPC were grouped together (meaning internal controls included at same location). It also may help with interpretation.

Eg. In Fig 6 it seems that CD34+ HSPC from Patient 3 (P3) responded well to; F+A treatment when co-cultured with auto (P3) MSC (shown in Fig 6A left column) and somewhat responding to (P25) MSC (shown in Fig 6C near left column), however did not respond to (P31) MSC after F+A treatment. Is this correct? Does this mean (P31) MSC are always non-responders to F+A?

Likewise is it correct that HSPC from patient (P11) ; do not respond to auto (P11) MSC after F+A, but do respond to treated (P25) MSC (Fig 6B left col) and somewhat respond to (P31) MSC (FIG 6A right column)? Overall suggesting that the MSC from this patient do not respond to F+A treatment?

Could the authors also please give their criteria for choosing each of the HPSC and MSC groups represented. Were these solely selected on availability of enough cells or selected more on whether the data fit with hypothesis. Similar comment for Figure 6.

Authors' answer: We apologize that our submitted figure structures were difficult to follow. We made the recommended changes on the X-axes and also added additional explanations on the top of each panel to clearly define whether autologous or allogeneous settings were used. To this end, new Figure 6a presents standard autologous co-cultures of MSCs and HSPCs from n=6 patients (P3, P7, P6, P8, P9 and P11). All of these patients were defined as erythroid P+A responders in new Figure 2e. New Figure 6a serves to highlight that erythroid response of HSPCs is boosted when they are co-cultured with MSCs, while no or much lower erythroid differentiation is observed when HSPCs alone are treated with P+A (monoculture = HSPCs alone). New Figure 6b and 6c further highlight the critical importance of MSCs in P+A induced erythroid differentiation of HSPCs.

However, in these experiments allogeneous settings were used. For instance, new Figure 6b shows that MSCs from patients defined as P+A responders in new Figure 2e (P11, P9, P6 and P3) may boost erythroid differentiation in HSPCs that were non-responders in autologous co-cultures (P25, P29, P23 and P31). Autologous co-cultures on new Figure 6b are shown in blue. In contrast to new Figure 6b, new Figure 6c presents that MSCs from patients categorized as P+A non-responders impair erythroid differentiation of HSPCs that were categorized as responders to P+A in autologous co-cultures. Overall, this figure shows that the source of MSCs is critical to control erythroid differentiation of HSPCs under treatment with P+A. Even in previously non-responsive HSPCs, erythroid differentiation may be boosted when MSCs of P+A responders are selected for co-cultures. And opposite, even in good responding HSPCs erythroid differentiation can be impaired when non-responding MSCs are used. Overall, our data are in line with previously published studies that propose MSCs as therapeutic targets in myelodysplastic malignancies (Poon et al, *Leukemia*. 2019 Jun;33(6):1487-1500. doi: 10.1038/s41375-018-0310-y; Wobus et al, *Leukemia*. 2021 Oct;35(10):2936-2947. doi: 10.1038/s41375-021-01275-5.). We further elaborate on this phenomenon in new Figures 7 and 8 and hypothesize, that the MSCs from responders change ECM properties (e.g. morphology, ligand composition) in response to PXS-5505. This then allows improved erythroid differentiation of HSPCs due to the changes in ECM sensing by integrins. The explanation of this data were provided in our initially submitted manuscript in the paragraph "*Induction of erythroid differentiation by P+A treatment is BM stroma-dependent*" (current version of manuscript: pages 10-11).

The criteria to for choosing MSCs and HSPCs were the following:

New Figure 6a: only P+A responders in autologous co-cultures were selected in order to demonstrate that the presence of MSCs is required for the boost of erythropoiesis. Here, we could not use non-responder patients since testing this hypothesis in non-responders is not possible (no P+A effects on erythropoiesis).

New Figure 6b: we selected MSCs from n=4 P+A responders and HSPCs from n=4 P+A non-responders to test the hypothesis that responder MSCs will rescue erythroid differentiation in non-responsive HSPCs. The number of patients was restricted by the availability of primary material.

New Figure 6c: we selected MSCs from n=4 P+A non-responders and HSPCs from n=3 P+A responders to test the hypothesis that non-responder MSCs could inhibit erythroid differentiation in responsive HSPCs. The number of patients was restricted by the availability of primary material.

Reviewer #1:

Figure 7. examines impact of P+A treatment on ECM stiffness, collagen (Fig 6 A-E) and impact on erythroid % change in alone MSC, PXS-5505 treated MSC or with HSC co-culture HSPC for P3, P10 and P11 MSC after decellularisation (possibly data from same P3, P10, P11 experiment as shown in Fig 5) with readout being CFU-E assay instead in Fig 7F-J.

The colony assay readouts only include % CD235a+ 45- cells weight no reference to total number of colonies produced or HSPC survival. It would be ideal to show all readouts including total CFU numbers for each group for each treatment.

As several controls may not be included (only groups shown HSC separated by transwells it is hard to fully interpret data. Each treatment group could have matched untreated control.

To aid appreciation of the data it would be helpful if Fig 7 F-H were changed so the top tier contains data only from HSC+MSC co-cultured cells (including all treatment comparisons) then the bottom tier of figures be the matched HSC + decellularized ECM datas. This would help enormously with interpretation. At present the data seems to suggest that CD34+ cultured alone show superior differentiation to CD235a+45- cells (%) than PSX-5505 treated alone ECM. Is this correct interpretation (maybe easier once figure if re-organised) and please explain if so.

Author comment: Thanks for the reviewer's comment regarding potentially missing controls and data interpretation. We repeated transwell experiments for initially submitted Figure 7B by using samples of P3, P6 and P9 and now provide additional untreated controls in new Supplementary Figure 16a. In addition, we now show new flow cytometry data (new Supplementary Figure 16a) in parallel with erythroid and myeloid colony count data (new Supplementary Figure 16b, c). Of note, quantitative flow cytometry data and manual erythroid colony count data show very consistent and comparable results. This was the reason for us to use flow cytometry assessments as an objective quantitative assessment of erythropoiesis in most of the experiments instead of laborious manual colony count. In addition, new Supplementary Figure 16d (*Results*: page 11, line 265-266) shows recovery of live HSPCs after 4 days of MSC/HSPC co-culture demonstrating that comparable numbers of live HSPCs were available for colony assays in all treatment conditions. As shown in new Supplementary Figure 2b, non-toxic doses of PXS-5505 (2 μ M) and 5-AZA (0.1 μ M) were administrated in our co-cultures. We therefore observed similar HSPC count after treatment as compared to untreated controls.

We would also like to clarify that in our manuscript each MN patient and HY control used for in vitro co-culture or in vivo xenograft studies has its unique identifier (P1, P2...P33 and HY1, HY2...HY7). This number identifies the same patient or healthy control in all our experiments and figure panels. For example P3 in new Figure 7b is the same patient as in new Figure 5d. These unique identifiers were also clearly defined in Table 1 (patient characteristics) in our initial submission. This table also shows how the samples were used in the study (for example, in vitro studies, in vivo PDX experiments or both). It is also important to indicate that we could not include all treatment controls in each functional experiment shown in Figure 7 due to the very limited amount of primary HSPCs. However, for every patient presented in Figure 7 complete sets of controls are actually available in the experiments for new Figure 2e (heatmap overview) and new Supplementary Figure 4a (individual data points). We could not repeat all control arms again in the experiments shown in new Figure 7f-h (original Figure 7F-H) due to the limited availability of primary material. Therefore, we focused on critical controls in order to:

1. demonstrate that the effect of P+A combination is reproducible in multiple experiments and
2. set a control arm for functional studies. For example, we included P+A treated MSC/HSPC cell contact co-culture in new Figure 7f-h since this is a critical control to demonstrate that the experiment is reproducible and provide positive control for comparisons with the effects observed on the decellularized ECM. All other treatment arms and controls for MSC/HSPC cell contact co-culture can be found in new Supplementary Figure 4a.

The interpretation of the experiments was provided in the text of the original submission (in the current version of manuscript: page 12, line 274-280): “we next cultured MN HSPCs of n=3 P+A erythroid responders on decellularized ECM of either vehicle or PXS-5505 pre-treated autologous MSCs (Fig. 7f-h). Two of these patients (P3 and P11) were erythroid responders in both in-vitro co-culture and in the in-vivo xenografted model. Of note, HSPCs of these two patients showed remarkable erythroid response after P+A treatment on decellularized ECM of PXS-5505-treated MSCs (PXS-5505 ECM), although at slightly lower levels as compared to co-culture with viable MSCs (Fig. 7f, g)”.

But we would also like to clarify that the initially submitted Figure 7F-G (new Figure 7f, g) showed that after P+A treatment HSPCs cultured on PXS-5505 ECM showed increased CD235a+CD45-cell production as compared to HSPCs monoculture. In addition, we decided that controls of HSPCs alone, which were shown in the initially submitted Figure 7F-H (last two columns of each panel, shown in green) are not suitable controls for the experiments on the decellularized ECM. This is due to the fact that the culture of HSPCs on ECM may further modulate erythroid differentiation when compared to the culture of HSPCs directly on the polystyrene surface. This could confuse and create difficulties in the data interpretation for the reader. Since HSPCs differentiation on the PXS-5505 treated ECM is best controlled by untreated ECM, we decided to remove HSPCs alone data from initially submitted Figure 7F-H. Revised data are presented in new Figure 7f-h.

Reviewer #1:

Figure 8 is mechanistic studies using blocking antibodies to proposed integrin avb3 interaction and signalling pathways. Only impact on % CD235a, CD45- cells shown. No data on overall HPSC survival in each of these cultures is shown, nor impacts of treatments on HSPC differentiation to other cell populations. integrin avb3 is not a completely novel target. Ideally blocking this integrin in mouse studies in vivo would be gold standard.

Authors' answer: In response to the reviewers request we now provide additional data on the myeloid cell differentiation (CD45+CD33+) (new Supplementary Figure 19a). An additional explanation was added to the manuscript text, page 13. Moreover, in new Supplementary Figure 19b we show recovery of live HSPCs after 4 days of co-culture, which is comparable in all conditions regardless of the used inhibitor. Indeed, the concentrations of all inhibitors in our study were optimized to be within non-toxic range (data provided in new Supplementary Figure 18b). We also would like to mention that the used concentrations of blebbistatin and Y27632 were comparable or lower than in previously published reports (Choi et al, Sci Adv. 2017 Jan 6;3(1):e1600455. doi: 10.1126/sciadv.1600455; Shin et al, Cell Stem Cell. 2014 Jan 2;14(1):81-93. doi: 10.1016/j.stem.2013.10.009). We thank the reviewer for suggesting integrin blocking experiments in vivo. However, this experiment in NSG PDX model is technically difficult due to poor erythroid differentiation of human HSPCs in NSG mice. In our PDX models, only 2 patient HSPCs samples were able to differentiate into erythroid progenitors in mouse bone marrow. HSPCs of other patients were able to differentiate into erythroid cells only in ex vivo erythropoietin-supplemented assays (new Figure 5 of our manuscript). Since primary material of those two patients is not available anymore, new patient samples need to be screened in vivo in order to find samples, which are able to differentiate in erythroid cells in the mouse bone marrow. We hope that this interesting experiment will be possible in the future in NUP98-HOXD13 mice. Currently, we are working to show that the effects of P+A can be reproduced in this model (preliminary data are presented above).

Reviewer #2 (Remarks to the Author); expert in MPN:

This is an interesting and thorough study by Xu et al. that addresses the mechanism by which

combination treatment with lysyl oxidase and 5-AZA treatment functions in myeloid neoplasms. Overall this study is novel and timely, and the dissection of the cellular and molecular requirements for this combination treatment makes this work even stronger. I have several comments that I believe could strengthen the manuscript:

1. Figure 1: I understand the authors are investigating myeloid neoplasms as a whole here, but I think it would be helpful to see LOXL2 expression, concentration, and activity separated by disease subtype. For instance, in figure 1A, the error bar is quite big for the LOXL2 gene expression in MSCs. Can the authors show LOXL2 mRNA expression in MSCs for MDS versus CMML etc? Also, if possible, the authors should add more samples here (3-5 more would suffice) to strengthen the conclusion that LOXL2 is up-regulated in MSCs from MN patients.

Authors' answer: We thank the reviewer for the overall positive evaluation of our work and thank you for the helpful suggestions for improvement. In response to this comment we have added more samples in order to be able to improve Figure 1A to show gene expression levels of LOX/LOXL separated by MDS versus CMML as suggested by the reviewer (new Figure 1a). A further detailed breakdown of the data – also including other myeloid subentities for which we did not have high sample numbers is depicted in new Supplementary Figure 1a, b (*Results*: page 5, line 96-101) and Supplementary Table 1. Because the data for enzymatic activity and concentration is highly enriched for MDS cases, we also opted to address this comment by depicting the separated data in new Supplementary Figure 1e instead of amending the main figure.

Reviewer #2

2. Figures 1H and I: the authors should include healthy donors in these viability curves, and compare the response of patient cells to healthy donors. Additionally, the authors should include at least 2 patients per type of disease.

Authors' answer: As suggested by the reviewer we included an additional n=3 healthy controls (HY5-HY7) in the viability assessments and expanded the samples size to at least 2 patients per disease type (new Supplementary Table 4; *Results*: page 6, line 128). New Figures 1h, i were amended accordingly to include the additional data. The data are broken down further in new Supplementary Figure 2a-c. There was no difference in the response between healthy control and patient samples. It should be noted that in this study we selected non-toxic concentrations of both agents since the primary goal was to restore differentiation rather than eliminate neoplastic cells by cytotoxic effects.

Reviewer #2

3. Figure 2A: The authors show in Figure 1 that PXS treatment leads to decreased collagen crosslinking in MDS patients with bone marrow fibrosis. In the co-culture shown in 2A, the authors should stain MSCs for reticulin (or even collagen expression) to determine whether PXS decreases collagen deposition in bone marrow stromal cells.

Authors' answer: In accordance with the reviewer's suggestion we quantitatively assessed cross-linked collagen deposition by MSCs after MSC/HSPC co-cultures (n=3). We preferred collagen assessment method based on Sircol™ kit (Biocolor) over reticulin staining due to possibility for

precise quantitative colorimetry-based assessment of a collagen deposition (*Supplementary Methods*: pages 3-4). The results are now shown in new Supplementary Figure 2f. Indeed, PXS-5505 strongly inhibited cross-linked collagen deposition in co-cultures. Importantly, this effect was mediated by PXS-5505, but not 5-AZA. An additional text passage was added on the page 6 (line 130-133) of the revised manuscript (Results section).

Reviewer #2

4. Figure 4: in this mouse model, do mice develop bone marrow fibrosis? If so, does this combo treatment prevent or reverse fibrosis in addition to its effects on erythroid differentiation? If so, the authors should highlight this finding. If this mouse model does develop BM fibrosis but the combo treatment doesn't affect it, the authors should comment on this.

Authors' answer: In our study, bone marrow fibrosis and hyperosteogeny were clearly detectable in the xenografts of the patient with PMF grade 3 fibrosis (P32). To highlight this we have enlarged the corresponding images in new Figure 4g, Gomori reticulin staining. From that data we can conclude that PXS-5505 treatment alone had a strong inhibitory effect on the formation of reticulin fibers and osteogeny despite comparably high engraftment (shown by parallel human mitochondria staining). We can therefore say PXS-5505 and P+A are able to “prevent” or “contain” fibrosis. Whether the substance / combination is also actually able to “reverse” fibrosis cannot be evidenced by our data because we don't have baseline values before the start of treatment. 5-AZA treatment alone also strongly inhibited fibrosis in this patient case. However, 5-AZA anti-fibrotic activity was mainly attributed to its ability to reduce human engraftment (compare human mitochondria staining for PXS-5505 and 5-AZA in new Figure 4g, h). Nearly complete absence of reticulin fibers was observed in P+A treated mice (new Figure 4g, h). We amended the text in the manuscript to highlight these observations (Text passage marked in yellow on page 9, line 205-212 in the section of “*Results*”).

Reviewer #2

5. Figure 7: If PXS treatment increases collagen I expression in MCS, how do the authors reconcile this with their claim that PXS treatment reduces MN disease burden? Increased MSC collagen expression seems to me an indication of increased MN disease burden. The authors should comment on this in the discussion.

Authors answer: We thank the reviewer for this comment. We do not think that PXS-5505 increases total collagen expression, but rather mediates re-distribution and different assembly of collagen fibers. In detail, our mass spectrometry data (new Figure 7j) show that the increase in the collagen type I Alpha 2 chain (COL1A2) deposition is accompanied by decreased deposition of the collagen alpha-1(XI) chain (COBA1) and collagen-interacting protein lumican (LUM), both of which also play essential roles in extracellular matrix organization and tumor microenvironment. Previous studies showed that increased deposition of COBA1 and LUM both contributed to tumor invasion and progression in certain tumors (Liu et al, *FASEB J.* 2021 Jun;35(6):e21603. doi: 10.1096/fj.202100054RR.; Brézillon et al, *FEBS J.* 2013 May;280(10):2369-81. doi: 10.1111/febs.12210.). Moreover, our phase contrast microscopy images of PXS-5505 treated decellularized ECM show clear re-organization of collagen fibers (new Figure 7d and new Supplementary Figure 18a for case P3). For this revision, we additionally provide novel electron microscopy images that show re-distribution of the ECM fibers in PXS-5505 treated ECM of MDS patient, which more resemble healthy ECM (new Supplementary Figure 17). We commented on the

observed modifications of PXS-5505 treated ECM in the “*Discussion*” section of the revised manuscript (page 14).

Reviewer #3 (Remarks to the Author); expert in extracellular matrix:

This is a wide ranging and detailed pre-clinical study using patient-derived bone marrow cells to investigate synergy between azacytidine and the pan-lysyl oxidase inhibitor PXS-5505 to ultimately potentially improve clinical outcomes in myelodysplastic diseases. The manuscript is clearly written, though this reviewer would have preferred fewer abbreviations. The use of PXS-5505 instead of the typical BAPN is a strength in light of the relative toxicity profiles of the two inhibitors, and potential clinical translation. The comprehensive nature of the assays performed in vitro and in xenografts is to be commended, as is transparency of data presentation. Below are some comments/questions for the authors to consider in a (most likely) minor revision.

Authors’ answer: We thank the reviewer for the overall positive evaluation of our work and for the constructive and helpful suggestions for improvement.

Reviewer #3

1. Why were only 4 healthy control subjects enrolled in this study? Does this provide sufficient data to make firm conclusions?

Authors answer: In response to this comment, we have increased the number of healthy control samples by n=3 to add up to a total of n=7 (new Table 1; new Figure 1e, h; new Supplementary Figure 4d, 5 and 7d). Results were identical to those of the initial 4 samples, i.e. none of the treatment conditions significantly changed hematopoietic differentiation in healthy samples.

In general, we did not focus on treating healthy samples with 5-AZA or PXS-5505+5-AZA (P+A) because treatment of healthy hematopoiesis for the purpose of hematologic improvement is not highly relevant for the translation of our novel drug combination (P+A) into the clinic. We thoroughly controlled all experiments with non-treatment controls and added these within the revision to experiments where they were missing.

The question is, which parameters should be controlled by additionally including healthy cells on top of the non-treatment controls? From our perspective we included healthy controls in the in-vitro experiments because we wanted to check whether P+A treatment could possibly even further increase erythropoiesis in healthy hematopoiesis (e.g. similar as erythropoietin). This didn’t seem to be the case. Another question, which we think is important, is hematotoxicity e.g. on healthy hematopoiesis. To this end, we have now supplied comprehensive health data on the mice in our PDX experiments, novel hematopoiesis data on long term exposure of rats to PXS-5505 and referral to early phase clinical data in humans. All of these data demonstrate that PXS-5505 seems to have a highly favorable (almost absent) toxicity profile. The toxicity profile of 5-AZA on the other hand is well known due to its > 10-years usage in clinical practice and consists of anemia, neutropenia and thrombocytopenia. To this end, in all of our experiments, our novel drug combination was either beneficial or at least neutral on hematopoietic differentiation and output as compared to the single substances or non-treatment controls.

Reviewer #3

2. Was LOX family enzyme activity determined in Azacytidine-treated only MN bone marrow plasma cells compared to non-treated bone marrow plasma cells? Azacytidine is known to up-regulate LOX family members in some, but not all tumor cells. If Azacytidine upregulation of LOXs occurs in MN, an explanation regarding why the combination of azacytidine + PXS-5505 results in a better outcome could be that PXS-5505 may prevent lysyl oxidase-driven fibrosis that is otherwise stimulated by Azacytidine.

Authors' answer: Thanks to the reviewer for this interesting comment. In the revised version of this manuscript, we assessed the effect of non-toxic doses of 5-AZA (0.1 μ M) on the LOX/LOXL activity in MN MSCs of n=5 patients (new Supplementary Figure 2g, new Supplementary Table 5). However, at least in the concentration used in the study (0.1 μ M) 5-AZA did not affect LOX/LOXL activity when compared to BAPN. Additional text describing this result was added on the page 6 (line 130-133) of the revised manuscript.

Reviewer #3

3. Figure 1C. Did lysyl oxidase enzyme activity assays include incubations +/- BAPN?

Authors answer: For the initially submitted Figure 1C, we did not include incubation with BAPN. Therefore, in accordance with the reviewer's comment, we repeated this experiment and set the background signal using BAPN treatment. The results shown in revised Figure 1c were improved as compared to our initially submitted data. Besides, signal-to-noise ratios were controlled by BAPN in activity assays presented in original Figure 1E and 1G (new Figure 1d-g) and Figure S1D (new Supplementary Figure 1c). This clarification was added in the *Supplementary Methods* (page 3) as well as new legends of new Figure 1c-g (text of manuscript: page 29) and new Supplementary Figure 1c, d. We also added a BAPN treatment arm in pan-LOX/LOXL activity inhibition assays (new Figure 1j and new Supplementary Figure 2g). These BAPN controlled assays readily confirmed significant increase in LOX and LOXL2 activity in MN patient samples.

REVIEWERS' COMMENTS

Reviewer #1 (Remarks to the Author):

This reviewer would like to thank the authors for their detailed respectful responses, their dedication in performing additional experiments to address reviewer comments and also in the provision of additional (for reviewer only) figures on authors data and current studies. This is very much appreciated.

Altogether the revised manuscript now reads well and together the newly re-arranged / generated data figures greatly enhance the overall readability and strength of this manuscript.

MINOR COMMENTS.

Figure 1. d,e compared to f,g.

These appear to be the identical data shown in two different ways. If so then maybe d,e are redundant as f, g gives the most information. If not then please explain the difference more clearly in legend. Also in the figure 1f,1g legend, please outline what the abbreviation 'BMF<1' and 'BMF>1' exactly means.

Results Figure 2.

The data shown suggesting that PXS-5505+A5-AZA treatment increases 'Erythropoietic response in autologous MSC co-cultured CD34+ cells are impressive. However all these data seem to be as percentages only. Total number of CFU generated per treatment group is not shown, only percentage that are erythroid. This raises the question whether these in vitro treatments may also have any impact on total haematopoietic cell number and total erythrocytes produced. Thus inclusion of a graph showing impact each of the respective treatments had on CD34+ cell expansion or total CFU-E or CFU-Myeloid generated in each condition would be a valuable addition. That is whether the strong increase in CFU-E or CD235a+ CD45- erythroid cells observed with F+A treatment in CD34+ from some patients actually came at the expense of reduced overall leukocyte production.

Results Figure 3.

Re. accuracy of Figure Title 'P+A facilitates erythroid differentiation of subclones with low mutational VAFs'. If this reviewer is correct in understanding the methodology was CD34+ cells co-cultured with autologous MSC for 7 days in presence of various treatments, then plated for CFU colony formation and the colony erythroid CD235a+ CD45- cells sorted for variant allele frequency. If method is correct then the data would suggest the P+A treatment seems to have only minor effect on VAF in the three patient samples shown, compared to untreated or 5-AZA alone treatment. In particular although P+A appears to reduce erythroid CEBPA VAF in patient 6 compared to untreated or 5-AZA alone (as highlighted in text), V+A treatment seems to have had the opposite effect in erythroid cells from patient P7 (almost 2 fold increased VAF for BCOR, DNMT3A and ASXL1 and IDH2 VAF with P+A treatment compared to 5-AZA treatment alone). If this interpretation is correct then maybe toning down the statement in figure 3 title and associated manuscript (p8) and discussion (p15 first paragraph) text would be appropriate (this also more closely reflects the data from a different model also showing no VAF differences with PXS-5505 in supp fig 15). However if interpretation is incorrect then please more clearly explain methodology (maybe as an expt outline graphic at top of Fig 3).

Figure 4. xenotransplant of patient bone marrow CD34+ (together with allogenic MSC) into NSG mice. Figure 4g Gomori (fibrosis) and human mitochondria stain on mouse femur sections. I appreciate these figures are now already enlarged, however to do this very interesting data justice would it be possible to include zoomed out images that cover a much larger portion of the femur as an additional supplementary figure. This would help readers gain appreciation of overall fibrotic distribution in xenotransplanted mouse femur (not only of a small selected magnified portion) and give more support for the statement (pg 9) that in PXS and PXS+AZA -treated xenotransplant mice there was strong inhibition of BM fibrosis.

Results section pg7 last paragraph. Imprecise wording.

Line 4 and 7. 'Of these, n=7 patients (27%) responded' sounds great, however would be more correct to be worded 'Of these, co-cultured CD34+ from n=7 patients ...responded'

Supp methods. Top re bone marrow derived MNC culture. Please indicate number of passages (eg. 1:4 spit) or duration that the bone marrow derived MSC were cultured. This would help the reader gauge potential amount of culture-associated epigenetic drift that may have occurred in the MSC used to generate the data in Figure 1,

Reviewer #2 (Remarks to the Author):

The authors have sufficiently addressed all my concerns. The manuscript has been substantially strengthened.

Reviewer #3 (Remarks to the Author):

The authors have addressed my comments very directly and very well, and appear to have responded thoroughly to the other reviewers' comments. This manuscript identifies a potential therapeutic effect of PXS-5505 for myelodysplastic conditions and will be of interest in this field.

Reviewer's Comments:

Reviewer #1 (Remarks to the Author)

This reviewer would like to thank the authors for their detailed respectful responses, their dedication in performing additional experiments to address reviewer comments and also in the provision of additional (for reviewer only) figures on authors data and current studies. This is very much appreciated.

Altogether the revised manuscript now reads well and together the newly re-arranged / generated data figures greatly enhance the overall readability and strength of this manuscript.

MINOR COMMENTS.

Reviewer 1

Figure 1. d,e compared to f,g.

These appear to be the identical data shown in two different ways. If so then maybe d,e are redundant as f, g gives the most information. If not then please explain the difference more clearly in legend. Also in the figure 1f,1g legend, please outline what the abbreviation 'BMF<1' and 'BMF>1' exactly means.

Authors' answer:

We agree with the reviewer that Figures 1f, g and 1d, e share the same source data set. However, we displayed it in these two different forms because figures d and e show the very clear total difference of increased LOXL2 concentration and activity as compared to healthy controls, while figures f and g focus on the dependency of these parameters on bone marrow fibrosis. In the latter two figures the total difference does not become as clear as in figures d and e, therefore, we suggest to keep both presentations of the data and outline this more clearly in the legend as suggested (legend of Figure 1, page 37). We also explained the abbreviation BMF (bone marrow fibrosis) in figure legends 1f and 1g (page 37).

Reviewer 1

Results Figure 2.

The data shown suggesting that PXS-5505+A5-AZA treatment increases 'Erythropoietic response in autologous MSC co-cultured CD34+ cells are impressive. However all these data seem to be as percentages only. Total number of CFU generated per treatment group is not shown, only percentage that are erythroid. This raises the question whether these in vitro treatments may also have any impact on total haematopoietic cell number and total erythrocytes produced. Thus inclusion of a graph showing impact each of the respective treatments had on CD34+ cell expansion or total CFU-E or CFU-Myeloid generated in each condition would be a valuable addition. That is whether the strong increase in CUF-E or CD235a+ CD45- erythroid cells observed with F+A treatment in CD34+ from some patients actually came at the expense of reduced overall leukocyte production.

Authors' answer:

We thank the reviewer for this further valuable suggestion for improvement. We have all the data on total colony counts from the experiments in Figure 2 available. As expected for MDS samples, the total numbers are heterogeneous on an individual patient level. However, this data very clearly rules out that "...the strong increase in CUF-E or CD235a+ CD45- erythroid cells observed with F(P)+A treatment in CD34+ from some patients actually came at the expense of reduced overall leukocyte production..." as apprehended by the reviewer. On the

contrary, although probably not statistically clear, there is more the impression that total hematopoietic production is increased by the P+A combination in some cases. We added this data in additional Supplementary Figure 6 showing that P+A does not significantly change total CFU assay outputs in P+A erythroid responder samples compared to untreated and single drug treated co-cultures. In addition to that CFU assay outputs of P+A responder patients contain significantly higher % of CFU-E colonies. An additional sentence explaining these results was added in the manuscript text, page 7.

Reviewer 1

Results Figure 3.

Re. accuracy of Figure Title 'P+A facilitates erythroid differentiation of subclones with low mutational VAFs'. If this reviewer is correct in understanding the methodology was CD34+ cells co-cultured with autologous MSC for 7 days in presence of various treatments, then plated for CFU colony formation and the colony erythroid CD235a+ CD45- cells sorted for variant allele frequency. If method is correct then the data would suggest the P+A treatment seems to have only minor effect on VAF in the three patient samples shown, compared to untreated or 5-AZA alone treatment. In particular although P+A appears to reduce erythroid CEBPA VAF in patient 6 compared to untreated or 5-AZA alone (as highlighted in text), V+A treatment seems to have had the opposite effect in erythroid cells from patient P7 (almost 2 fold increased VAR for BCOR, DNMT3A and ASXL1 and IDH2 VAF with P+A treatment compared to 5-AZA treatment alone). If this interpretation is correct then maybe toning down the statement in figure 3 title and associated manuscript (p8) and discussion (p15 first paragraph) text would be appropriate (this also more closely reflects the data from a different model also showing no VAF differences with PSX-5505 in supp fig 15). However if interpretation is incorrect then please more clearly explain methodology (maybe as an expt outline graphic at top of Fig 3).

Authors' answer:

We agree with the reviewer that although the VAFs of certain mutations markedly decreased after P+A treatment in some cases (e.g. CEBPA mutation in P6), but opposite effects could also be observed (e.g. BCOR mutation in P7). Therefore, we decided to follow the suggestion of the reviewer to remove the last two sentences of this results section in order to tone down this message (page 8). We also softened our conclusion in the Discussion section stating that P+A induced preferential expansion of "healthy" hematopoiesis (page 15). However, we would like to clarify that the main message of the data shown in Figure 3 is that the erythroid progenitors generated in the colony assay have significantly reduced mutational VAFs compared to the input MN HSPCs. This subclonal erythropoiesis is observed in all treatment arms as well as in the control arm. However, P+A further facilitates this subclonal erythropoiesis compared to other treatments. Therefore, we believe that the titles of Figure 3 ("P+A facilitates erythroid differentiation of subclones with low mutational VAFs") and the associated manuscript paragraph are appropriate.

Reviewer 1

Figure 4. xenotransplant of patient bone marrow CD34+ (together with allogenic MSC) into NSG mice. Figure 4g Gomori (fibrosis) and human mitochondria stain on mouse femur sections. I appreciate these figures are now already enlarged, however to do this very interesting data justice would it be possible to include zoomed out images that cover a much larger portion of the femur as an additional supplementary figure. This would help readers gain appreciation of overall fibrotic distribution in xenotransplanted mouse femur (not only of a small selected magnified portion) and give more support for the statement (pg 9) that in PXS and PXS+AZA -treated xenotransplant mice there was strong inhibition of BM fibrosis.

Authors' answer:

As per editorial suggestion to accommodate this point, we have uploaded all image source data for full public access at BioStudies EMBL-EBI portal (Accession code: S-BSST1021).

Reviewer 1

Results section pg7 last paragraph. Imprecise wording.

Line 4 and 7. 'Of these, n=7 patients (27%) responded' sounds great, however would be more correct to be worded 'Of these, co-cultured CD34+ from n=7 patients ...responded'

Authors' answer:

We thank the reviewer for the attentive reading of our manuscript. In accordance with this comment the word "patient" was changed to "the co-cultured HSPCs from" or "patient samples" in the mentioned parts of the manuscript (page 7).

Reviewer 1

Supp methods. Top re bone marrow derived MNC culture. Please indicate number of passages (eg. 1:4 spit) or duration that the bone marrow derived MSC were cultured. This would help the reader gauge potential amount of culture-associated epigenetic drift that may have occurred in the MSC used to generate the data in Figure 1,

Authors' answer:

In response to the reviewer's comment, we clarified that for in vitro and in vivo studies, expanded MSCs were used at passages 2 or 3. This information was added in the current section of Methods (page 17).

Reviewer #2 (Remarks to the Author)

The authors have sufficiently addressed all my concerns. The manuscript has been substantially strengthened.

Reviewer #3 (Remarks to the Author)

The authors have addressed my comments very directly and very well, and appear to have responded thoroughly to the other reviewers' comments. This manuscript identifies a potential therapeutic effect of PXS-5505 for myelodysplastic conditions and will be of interest in this field.